# Purinergic signaling in cochlear supporting cells reduces hair cell excitability by increasing the extracellular space

Travis A Babola[1], Calvin J Kersbergen[1], Han Chin Wang[1†], Dwight E Bergles[1,2,3*]

[1]The Solomon Snyder Department of Neuroscience, Johns Hopkins University, Baltimore, United States; [2]Department of Otolaryngology Head and Neck Surgery, Johns Hopkins University, Baltimore, United States; [3]Kavli Neuroscience Discovery Institute, Johns Hopkins University, Baltimore, United States

**Abstract** Neurons in developing sensory pathways exhibit spontaneous bursts of electrical activity that are critical for survival, maturation and circuit refinement. In the auditory system, intrinsically generated activity arises within the cochlea, but the molecular mechanisms that initiate this activity remain poorly understood. We show that burst firing of mouse inner hair cells prior to hearing onset requires P2RY1 autoreceptors expressed by inner supporting cells. P2RY1 activation triggers $K^+$ efflux and depolarization of hair cells, as well as osmotic shrinkage of supporting cells that dramatically increased the extracellular space and speed of $K^+$ redistribution. Pharmacological inhibition or genetic disruption of P2RY1 suppressed neuronal burst firing by reducing $K^+$ release, but unexpectedly enhanced their tonic firing, as water resorption by supporting cells reduced the extracellular space, leading to $K^+$ accumulation. These studies indicate that purinergic signaling in supporting cells regulates hair cell excitability by controlling the volume of the extracellular space.

*For correspondence:
dbergles@jhmi.edu

Present address: †Helen Wills Neuroscience Institute, University of California, Berkeley, Berkeley, United States

## Introduction

The developing nervous system must generate, organize, and refine billions of neurons and their connections. While molecular guidance cues forge globally precise neuronal connections between distant brain areas (*Dickson, 2002*; *Stoeckli, 2018*), the organization of local connections is initially coarse and imprecise (*Dhande et al., 2011*; *Kirkby et al., 2013*; *Sretavan and Shatz, 1986*). Coincident with the refinement of topographic maps, nascent circuits experience bursts of intrinsically generated activity that emerge before sensory systems are fully functional (*Kirkby et al., 2013*). This intrinsically generated activity consists of periodic bursts of high frequency firing that promotes the survival and maturation of neurons in sensory pathways (*Blankenship and Feller, 2010*; *Moody and Bosma, 2005*). The precise patterning of this electrical activity appears crucial for refinement of local connections, as its disruption results in improper formation of topographic maps (*Antón-Bolaños et al., 2019*; *Burbridge et al., 2014*; *Xu et al., 2011*) and impaired maturation and specification of sensory neurons (*Shrestha et al., 2018*; *Sun et al., 2018*). In all sensory systems that have been examined, spontaneous burst firing arises within their respective developing sensory organs (e.g. retina, olfactory neuroepithelium, and cochlea) (*Blankenship and Feller, 2010*; *Yu et al., 2004*). Although the mechanisms that induce spontaneous activity in the developing retina have been extensively explored, much less is known about the key steps involved in triggering auditory neuron burst firing in the developing cochlea. Understanding these processes may provide novel insights into the causes of developmental disorders of hearing, such as hypersensitivity to sounds

**eLife digest** As the brain develops, billions of cells respond to genetic and environmental cues to form the trillions of connections that make up its neural networks. However, before these brain circuits can respond to real life stimuli, their connections are refined by bursts of electrical activity. For example, sensory cells in the ear produce bursts of spontaneous electrical activity that mimic those made by sounds. This activity allows the neural network in the hearing system to 'practice' responding to sounds. However, the origin of these electrical bursts is unusual as they do not start in the sensory cells themselves, but are initiated by the non-sensory cells around them.

Past research has shown that as the ear develops these non-sensory cells, or supporting cells, release regular doses of a molecule called ATP. The supporting cells then detect their own ATP release using specialized receptor proteins on their surface. This self-stimulation causes the supporting cells to release potassium ions that interact with the sensory cells and trigger bursts of electrical activity. However, the identity of this ATP-detecting receptor was not known, and without this information it was unclear how the electrical activity starts and why it happens in rhythmic bursts.

To fill this knowledge gap, Babola et al. measured electrical activity in ear cells isolated from mice, and examined nerve cell activity in live mice during this critical stage of development. This revealed that the bursts of activity in the ear depend on a receptor called P2RY1 which can be found on the supporting cells located next to sensory cells. When P2RY1 is activated it triggers the release of calcium ions inside the supporting cells. This opens channels in the cell membrane, allowing the potassium ions to flow out and electrically activate the sensory cells.

But, when the potassium ions leave the supporting cells, water is drawn out with them, causing the cells to shrink and the space around the cells to get bigger. As a result, the released potassium ions disperse more quickly, moving away from the sensory cells and stopping the burst in electrical activity. Conversely, when P2RY1 is inhibited, this causes the supporting cells to swell, trapping potassium ions near the sensory cells and making them fire continuously. This indicates that bursts in electrical activity are controlled by the rhythmic swelling and shrinking of supporting cells.

Although supporting cells cannot detect sound themselves, they seem to play a crucial role in developing the hearing system. A better understanding of these cells could therefore aid research into hearing problems without a known cause such as hypersensitivity to sound, tinnitus, and complex auditory processing disorders in children.

and auditory processing disorders that prevent children from communicating and learning effectively.

The mechanisms responsible for initiating spontaneous activity appear to be unique to each sensory system, reflecting adaptations to the structure and cellular composition of the sensory organs. In the cochlea, two distinct models have been proposed to initiate burst firing of inner hair cells (IHCs). One model proposes that the initiation of burst firing results from intermittent hyperpolarization of tonically active IHCs by cholinergic efferents (*Johnson et al., 2011*; *Wang and Bergles, 2015*), which provide prominent inhibitory input to IHCs prior to hearing onset (*Glowatzki and Fuchs, 2000*). While transient activation of acetylcholine receptors in acutely isolated cochleae caused IHCs to switch from sustained to burst firing (*Johnson et al., 2011*), in vivo recordings from auditory brainstem revealed that neuronal burst firing remains, with altered temporal structure, in α9 acetylcholine receptor knockout (KO) mice (*Clause et al., 2014*) that lack functional efferent signaling in IHCs (*Clause et al., 2014*; *Johnson et al., 2013*). Burst firing also persists in IHCs and auditory neurons in cochleae maintained in vitro without functional efferents (*Johnson et al., 2013*; *Tritsch et al., 2007*). These findings suggest that cholinergic efferents modulate the temporal characteristics of bursts, but are not essential to initiate each event.

An alternative model proposes that IHCs are induced to fire bursts of action potentials by the release of $K^+$ from nearby inner supporting cells (ISCs), which together form a transient structure known as Köllikers organ (Greater Epithelial Ridge) that is prominent in the cochlea prior to hearing onset. $K^+$ release from ISCs occurs following a cascade of events that begins with the spontaneous release of ATP and activation of purinergic autoreceptors (*Tritsch et al., 2010a*). Purinergic receptor

activation induces an increase in intracellular $Ca^{2+}$ in ISCs, opening of $Ca^{2+}$-activated $Cl^-$ channels (TMEM16A), efflux of $Cl^-$ and subsequently $K^+$ to balance charge (*Tritsch et al., 2007*; *Wang et al., 2015*). The loss of ions during each event draws water out of ISCs through osmosis, leading to pronounced shrinkage (crenation) of ISCs. While these pathways have been extensively studied in vitro, the molecular identity of the purinergic receptors has remained elusive and few manipulations of this pathway have been performed in vivo, limiting our understanding of how spontaneous activity in the cochlea influences patterns of neuronal activity in auditory centers at this critical stage of development.

Here, we show that the key initial step in generation of spontaneous activity in the auditory system involves activation of P2RY1 autoreceptors in ISCs. These metabotropic receptors induce $Ca^{2+}$ release from intracellular stores that allow TMEM16A channels to open. Pharmacological inhibition of P2RY1 or genetic deletion of *P2ry1* dramatically reduced burst firing in spiral ganglion neurons (SGNs) and blocked the coordinated, spatially restricted activation of ISCs, IHCs, and SGNs in the cochlea. Unexpectedly, P2RY1 activation also promoted the dissipation of $K^+$ away from IHCs by increasing the volume of extracellular space. Conversely, inhibition of P2RY1 reduced the extracellular space and restricted the redistribution of $K^+$ within the cochlear epithelium, causing IHCs to depolarize and fire tonically, demonstrating an important role for purinergic receptor-mediated extracellular space changes in controlling IHC excitability. Using in vivo widefield epifluorescence imaging of the auditory midbrain in unanesthetized mice, we show that acute inhibition of P2Y1 dramatically reduced burst firing of auditory neurons in isofrequency domains. Together, these data indicate P2RY1 autoreceptors in non-sensory supporting cells in the cochlea play a crucial role in generating bursts of activity among neurons that will ultimately process similar frequencies of sound, providing the means to initiate the maturation of auditory pathways before hearing onset.

## Results

### Supporting cell spontaneous currents require calcium release from intracellular stores

Periodic release of ATP from ISCs in the developing cochlea initiates a signaling cascade in these cells that increases intracellular calcium ($Ca^{2+}$), opens $Ca^{2+}$-activated $Cl^-$ channels (TMEM16A), and ultimately results in efflux of chloride and $K^+$ into the extracellular space. Although the increase in intracellular $Ca^{2+}$ following activation of purinergic autoreceptors is sufficient to induce both depolarization and osmotic shrinkage (*Wang et al., 2015*), the relative contributions of $Ca^{2+}$ influx (e.g. through $Ca^{2+}$-permeable, ionotropic P2X receptors) and release from intracellular stores (e.g. following metabotropic P2Y receptor activation) to these cytosolic $Ca^{2+}$ transients is unclear. To define the signaling pathways engaged by purinergic receptor activation, we examined the sensitivity of spontaneous ISC whole-cell currents and crenations to inhibitors of intracellular $Ca^{2+}$ release pathways (*Figure 1A*). Spontaneous inward currents and crenations were abolished following a 15 min incubation of excised cochlea in BAPTA-AM (100 µM), a cell permeant $Ca^{2+}$ chelator (*Figure 1B–F*), and after depleting intracellular $Ca^{2+}$ stores with thapsigargin (2 µM), an inhibitor of endoplasmic reticulum $Ca^{2+}$-ATPase (*Figure 1B–F*). These data suggest that $Ca^{2+}$ release from intracellular stores is necessary for spontaneous electrical activity in ISCs.

Metabotropic $G_q$-coupled receptors typically induce PLC-mediated cleavage of phosphatidylinositol 4,5-bisphosphate (PIP2) and subsequent binding of inositol trisphosphate ($IP_3$) to $IP_3$ receptor-channels on the endoplasmic reticulum to release $Ca^{2+}$ into the cytoplasm. To determine if PLC signaling is necessary to generate spontaneous activity in ISCs, we recorded spontaneous currents and crenations from ISCs in the presence of U73122 (10 µM), a PLC inhibitor, and U73343 (10 µM), an inactive succinimide analog. The frequency of spontaneous currents and crenations were significantly reduced by U73122, but not by U73343 (*Figure 1B–F*); the amplitudes and charge transfer of residual activity also trended lower during PLC inhibition, but this did not reach significance due to high variance in the sizes of the spontaneous responses (*Figure 1B–F*). Intracellular $Ca^{2+}$ stores can also be mobilized through calcium-induced calcium release (CICR) involving ryanodine receptors. However, neither spontaneous currents nor crenations were affected by ryanodine (10 µM) (*Figure 1—figure supplement 1*). Together, these results suggest that engagement of a $G_q$-coupled purinergic

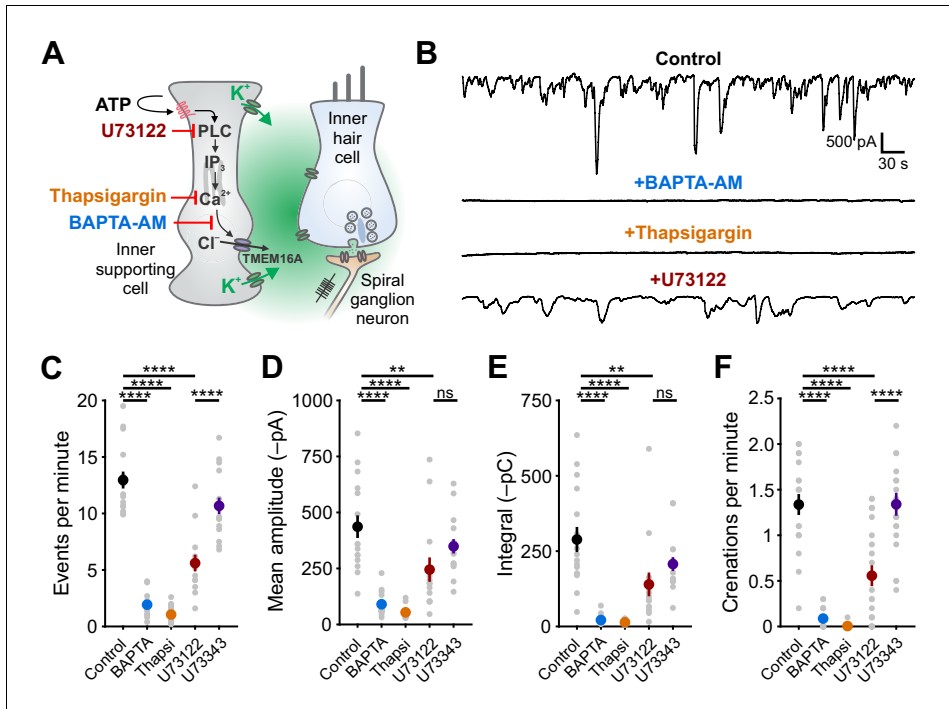

**Figure 1.** Ca²⁺ release from intracellular stores is required for spontaneous currents and crenation in inner supporting cells. (**A**) Model of ATP-mediated depolarization of inner hair cells. ATP: adenosine triphosphate, PLC: phospholipase C, IP₃: inositol triphosphate, TMEM16A: transmembrane member 16A (Ca²⁺-activated Cl⁻ channel). Inhibitors of key steps in this pathway are indicated. (**B**) Whole-cell voltage-clamp recordings from inner supporting cells after pre-incubating with indicated inhibitors. (**C**) Quantification of ISC spontaneous current frequency in the presence of the indicated inhibitors. Data shown as mean ± SEM. n = 16 cells, 11 cochleae from six postnatal day (P) 6–8 rats (control), 18 cells, 10 cochleae from five rats (BAPTA-AM; 100 µM), 20 cells, 11 cochleae from six rats (Thapsigargin; 2 µM), 14 cells, 11 cochleae from seven rats, (U73122; 10 µM), and 16 cells, 10 cochleae from five rats (U73343; 10 µM). ****p<5e-5, one-way ANOVA. (**D**) Quantification of ISC spontaneous current amplitude in the presence of indicated inhibitors. Data shown as mean ± SEM. n values are reported in (**C**) (one-way ANOVA; ****p<5e-5, **p<0.005, ns: not significant). (**E**) Quantification of ISC spontaneous current charge transfer (integral) in the presence of indicated inhibitors. Data shown as mean ± SEM. n values are reported in (**C**) (one-way ANOVA; ****p<5e-5, **p<0.005, ns: not significant). (**F**) Quantification of ISC crenation (cell shrinkage) frequency in the presence of indicated inhibitors. Data shown as mean ± SEM. n = 19 videos, 11 cochleae from six rats (control), 15 videos, 8 cochleae from four rats (BAPTA-AM), 22 videos, 12 cochleae from six rats (Thapsigargin), 23 videos, 17 cochleae from 10 rats (U73122), and 20 videos, 10 cochleae from five rats (U73343) (one-way ANOVA; ****p<5e-5, **p<0.005, ns: not significant). See *Figure 1—source data 1* for plotted values and statistics.

The online version of this article includes the following source data and figure supplement(s) for figure 1:

**Source data 1.** Plotted values and statistics for *Figure 1*.

**Figure supplement 1.** Inhibition of calcium-induced calcium release does not alter spontaneous currents or crenations.

**Figure supplement 1—source data 1.** Plotted values and statistics for *Figure 1—figure supplement 1*.

autoreceptor is a critical first step in initiating PLC-mediated Ca²⁺ release from intracellular stores and subsequent activation of TMEM16A channels.

## The metabotropic purinergic receptor P2Y1 is highly expressed by supporting cells

There are eight members of the metabotropic purinergic receptor family in mouse, four of which are G_q-coupled (P2RY1, P2RY2, P2RY4, and P2RY6). Gene expression studies in the developing mouse cochlea revealed that non-sensory cells express P2RY1 mRNA at high levels, >100 fold higher than any other P2RY (*Figure 2A*; *Scheffer et al., 2015*) and that expression of this receptor progressively increases during early postnatal development (*Figure 2A*, inset) concurrent with increases in

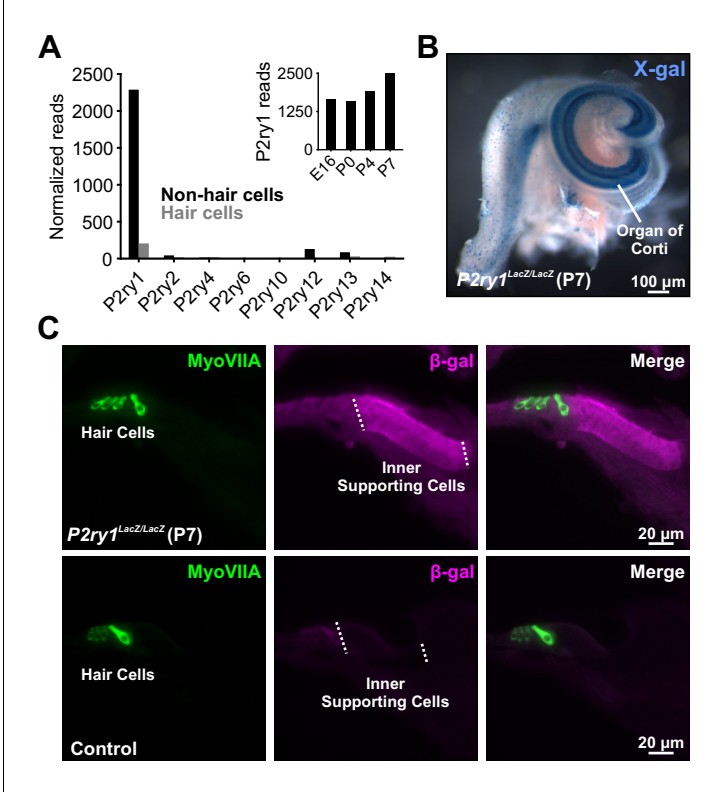

**Figure 2.** The metabotropic P2Y1 receptor is highly expressed by ISCs. (A) Expression levels of metabotropic purinergic receptors in hair cells (gray) and non-sensory cells (black) of the developing cochlea (postnatal day 7, P7). (inset) *P2ry1* expression in non-sensory cells over development. Data adapted from *Scheffer et al. (2015)*. (B) Image of a cochlea following X-gal reaction in *P2ry1* LacZ reporter mice. (C) Immunostaining for B-galactosidase in cochleae from P7 *P2ry1* LacZ (top) and control (bottom) cochlea.

spontaneous activity (*Tritsch and Bergles, 2010*). To determine which cells in the sensory epithelium express P2RY1, we isolated cochleae from *P2ry1-LacZ* reporter mice and performed X-gal staining. Intense blue labeling was present along the entire length of the cochlea within Kölliker's organ (Greater Epithelial Ridge; *Figure 2B*), and cross-sections of cochlea revealed that staining was present within ISCs, but not IHCs (Myosin VIIA, *Figure 2C*), indicating that P2RY1 is appropriately localized to sense ATP release from ISCs prior to hearing onset.

## P2RY1 signaling is required for spontaneous activity in ISCs and IHCs

To determine if P2RY1 is responsible for spontaneous ATP-mediated currents in ISCs, we examined the sensitivity of these responses and associated crenations to the P2RY1 selective antagonist MRS2500 (*Figure 3A,B*). Acute inhibition of P2RY1 with MRS2500 (1 µM) markedly reduced both spontaneous ISC currents (*Figure 3B,C*) and crenations (*Figure 3D,E*); near complete inhibition occurred within minutes at both room temperature (*Figure 3B,C*) and near physiological temperature (*Figure 3—figure supplement 1A–G*), with only sporadic, small amplitude events remaining that were not mediated by purinergic receptors (*Figure 3—figure supplement 1B–E*). Consistent with the involvement of P2RY1, the amplitude and total charge transfer of ISC events (*Figure 3—figure supplement 2A,B*) and size of spontaneous crenations (*Figure 3—figure supplement 2C,D*) were smaller in cochleae in *P2ry1* KO mice relative to controls. However, supporting cells in *P2ry1* KO mice exhibited some aberrant, gain-of-function activity consisting of more frequent, small amplitude currents (*Figure 3—figure supplement 2A,B*), that were not blocked by MRS2500 or broadspectrum P2 receptor antagonists (*Figure 3—figure supplement 2E,F*).

ATP-mediated signaling in ISCs activates TMEM16A, triggering $K^+$ efflux that depolarizes nearby IHCs. To assess whether P2RY1 signaling is also required for periodic excitation of IHCs prior to

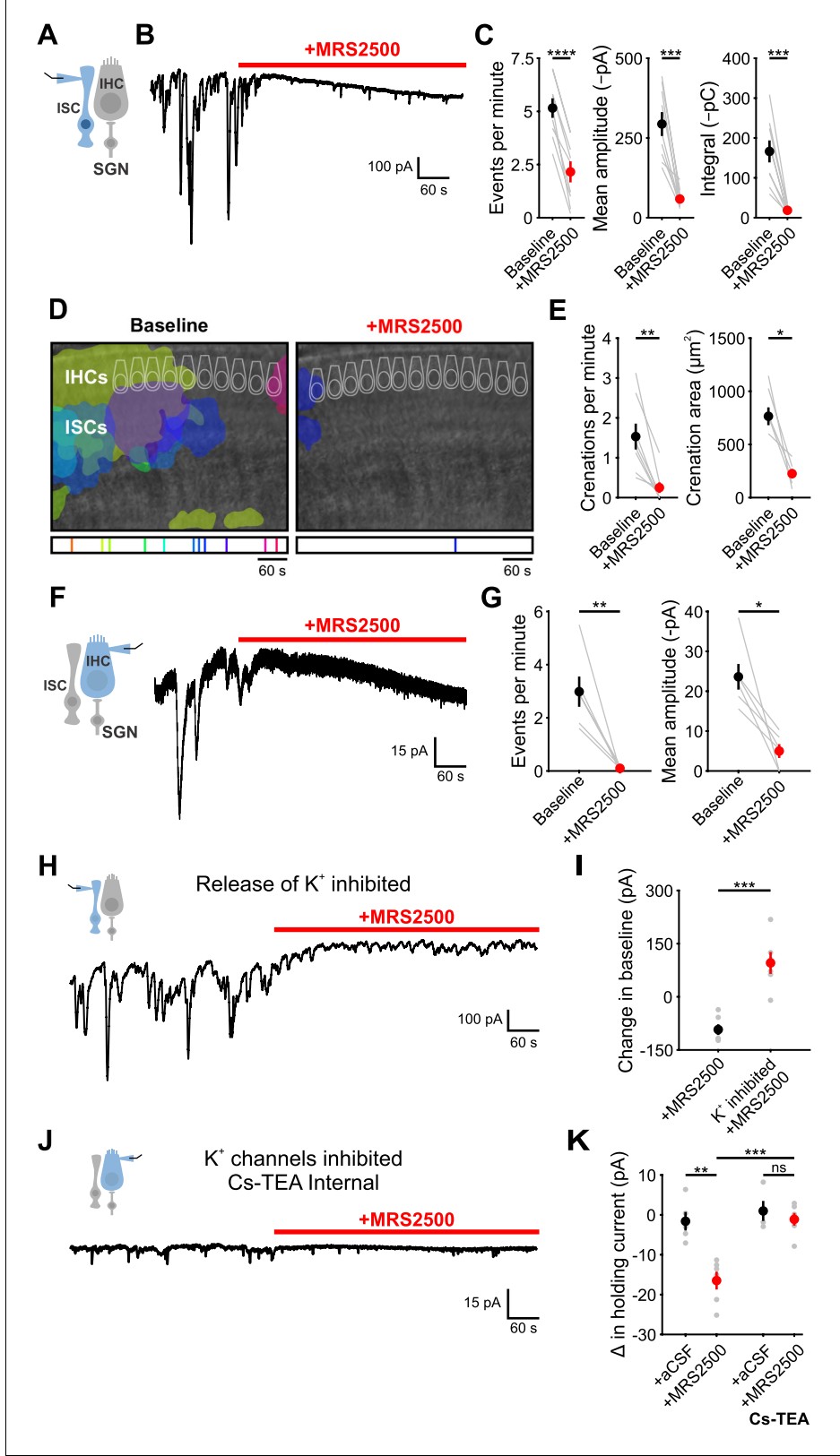

**Figure 3.** P2Y1 inhibition abolishes spontaneous currents in inner supporting cells and inner hair cells. (**A**) Schematic of whole-cell recording configuration from ISCs. (**B**) Spontaneous inward currents recorded from an inner supporting cell before and during application of MRS2500 (1 µM). Recordings were performed at room

*Figure 3 continued*

temperature (~25˚C). (C) Plot of event frequency, amplitude, and integral (charge transfer) before and after application of MRS2500. Measurement periods were five minutes long. n = 9 ISCs, 9 cochleae from 8 P6-8 mice (two-tailed paired Student's t test; ****p<5e-5, ***p<0.0005) (D) Intrinsic optical imaging performed before and after application of the P2RY1 antagonist, MRS2500 (1 µM). Detected crenations are outlined in colors based on time of occurrence as indicated by timeline below image. Imaging was performed at room temperature (~25˚C). (E) Plot of crenation frequency and area before and after application of MRS2500. n = 8 videos, 8 cochleae from 8 P6-8 mice (two-tailed paired Student's t test; **p<0.005) for frequency calculation and n = 5 cochleae (two-tailed paired Student's t test; *p<0.05) for area calculation. Cochleae that did not crenate after MRS2500 were excluded from the area calculation. (F) Schematic of whole-cell recording configuration from IHCs. (right) Whole-cell voltage clamp recording from an IHC before and during application of MRS2500. (G) Plots of event frequency and amplitude before and after application of MRS2500. n = 6 IHCs, 6 cochleae from 6 P6-8 mice (two-tailed paired Student's t test with Bonferroni correction; **p<0.005, *p<0.05). (H) Whole-cell voltage clamp recording of an ISC with application of MRS2500 following pre-incubation in aCSF containing $CdCl_2$ (100 µM), TTX (1 µM), ouabain (10 µM), and bumetanide (50 µM) to limit potassium release into the extracellular space. (I) Plot of the change in holding current, defined as the 95% percentile current value for each period. n = 6 ISCs, 6 cochleae from 6 P6-8 mice for each condition (two-tailed Student's t test; ***p<0.0005). (J) Whole-cell voltage clamp recording of an IHC with a Cs-TEA internal solution (to inhibit K+ channels) before and after MRS2500 application. (K) Plot of the change in IHC holding current following control (superfusion of aCSF only) and MRS2500 with K-MeS and Cs-TEA internal. n = 5 IHCs, 4 cochleae from 4 P6-8 mice for aCSF, n = 6 IHCs, 6 cochleae from six mice for MRS2500, n = 4 IHCs, 3 cochleae from three mice for aCSF with Cs-TEA internal, and n = 6 IHCs, 6 cochleae from six mice for MRS2500 with Cs-TEA internal (one-way ANOVA; ***p<0.005, **p<0.005, ns, not significant). See *Figure 3— source data 1* for plotted values and statistics.

The online version of this article includes the following source data and figure supplement(s) for figure 3:

**Source data 1.** Plotted values and statistics for *Figure 3*.
**Figure supplement 1.** P2ry1 inhibition abolishes spontaneous inward currents near physiological temperature.
**Figure supplement 1—source data 1.** Plotted values and statistics for *Figure 3—figure supplement 1*.
**Figure supplement 2.** Spontaneous inward currents and crenations are dramatically reduced in *P2ry1 KO* mice.
**Figure supplement 2—source data 1.** Plotted values and statistics for *Figure 3—figure supplement 2*.

hearing onset, we assessed the sensitivity of spontaneous IHC inward currents to MRS2500 (*Figure 3F*). Consistent with the supporting cell origin of IHC activity, application of MRS2500 (1 µM) also abolished spontaneous currents in IHCs (*Figure 3F,G*). Together, these data suggest that P2RY1 is the primary purinergic autoreceptor on ISCs responsible for inducing periodic excitation of IHCs prior to hearing onset.

## P2RY1 inhibition leads to extracellular K$^+$ accumulation

Although P2RY1 inhibition abolished most transient inward currents in both ISCs and IHCs, a progressively increasing inward current (downward shift in baseline) appeared in both cell types with prolonged application of MRS2500 (*Figure 3B,F*). Prior studies in CNS brain slices indicated that $G_q$-coupled purinergic receptors in astrocytes regulate extracellular K$^+$ concentration and neuronal excitability (*Wang et al., 2012*). The slowly progressing nature of the response in IHCs and ISCs suggest that it may arise from accumulation of K$^+$ released from cells in the organ of Corti. If this hypothesis is correct, then inhibiting the main sources of extracellular K$^+$ should diminish this inward current. Indeed, when IHC and SGN excitation was inhibited with cadmium ($CdCl_2$, 100 µM) and tetrodotoxin (TTX, 1 µM), and K$^+$ transporters Na,K-ATPase and NKCC were inhibited with ouabain (10 µM) and bumetanide (50 µM), no inward current was induced in ISCs upon blocking P2RY1 (*Figure 3H,I*). Similarly, if K$^+$ accumulation is responsible for the current in IHCs, it should be abolished when the ability of IHCs to detect changes in K$^+$ is reduced. When whole cell recordings were performed from IHCs using an internal solution containing Cs$^+$ and TEA, which blocks most IHC K$^+$ channels (*Kros et al., 1998*; *Marcotti et al., 2003*), MRS2500 also did not induce an inward current (*Figure 3J,K*). Together, these results suggest that P2RY1 has two distinct effects in the cochlea; it induces the transient inward currents that trigger IHC burst firing (*Figure 3F*; see also Figure 6D,G) and it accelerates clearance of K$^+$ away from IHCs within the organ of Corti.

To directly assess the relationship between P2RY1 activity and extracellular K$^+$ accumulation near IHCs, we monitored K$^+$ levels in the extracellular space using IHC K$^+$ channels. Focal P2RY1

stimulation with a selective agonist (MRS2365, 10 µM), which mimics the effect of endogenous ATP by eliciting an inward current and crenations in ISCs in control but not *P2ry1* KO mice (*Figure 4A–C*), was combined with assessments of the reversal potential of $K^+$ currents in IHCs using a voltage protocol similar to that used to assess extracellular $K^+$ buildup at vestibular calyceal synapses (*Lim et al., 2011*) (*Figure 4D–F*). This protocol consisted of: (1) a hyperpolarizing step to –110 mV to relieve $K^+$ channel inactivation, (2) a depolarizing step to +30 mV to activate outward $K^+$ currents, and (3) a step to –70 mV to obtain a 'tail' current. Because the conductance during this last step is largely mediated by $K^+$ channels, it is highly sensitive to shifts in $K^+$ driving force induced by changes in extracellular $K^+$ (*Contini et al., 2017*; *Lim et al., 2011*). Following transient stimulation of P2RY1, these $K^+$ tail currents immediately shifted inward, as would be expected if extracellular $K^+$ increases (*Figure 4G,J*), consistent with the effects metabotropic purinergic receptor stimulation on synaptically-evoked $K^+$ currents in IHCs (*Wang et al., 2015*). However, after a few seconds these $K^+$ currents shifted outward relative to baseline, indicative of a gradual decrease in extracellular $K^+$ below that present prior to P2RY1 stimulation, before gradually returning to pre-stimulation levels after several minutes (*Figure 4G,J*).

The outward shift in $K^+$ tail current closely followed the time course of supporting cell crenation (*Figure 4G*), suggesting that the shrinkage of ISCs induced by P2RY1 activation results in a prolonged increase in extracellular space around IHCs that allows greater dilution and more rapid redistribution of $K^+$ in the organ of Corti. Alternatively, buildup of extracellular $K^+$ alone may stimulate greater uptake. To determine if rapid increases in extracellular $K^+$ and $Cl^-$ were sufficient to stimulate $K^+$ redistribution in the absence of crenation, we focally applied KCl (130 mM) into the supporting cell syncytium near IHCs in the presence of P2RY1 antagonists (*Figure 4H,J*). As expected, this transient increase in extracellular $K^+$ induced an inward shift in $K^+$ tail currents and a brief optical change induced by fluid delivery; however, $K^+$ tail currents rapidly returned to baseline and did not shift outward, suggesting that extracellular $K^+$ (and $Cl^-$) elevation are not sufficient to enhance $K^+$ redistribution rates. In addition, we transiently stimulated P2RY1 in *Tecta-Cre;TMEM16A*fl/fl mice, in which purinergic receptor activation is preserved, but crenations are abolished (*Wang et al., 2015*). In these mice, ISCs failed to crenate, IHCs did not depolarize, and $K^+$ tail currents remained stable throughout the duration of the recording (*Figure 4I,J*). These results suggest that purinergic autoreceptors on ISCs influence extracellular $K^+$ levels by triggering $K^+$ release and by altering $K^+$ redistribution by controlling the size of the extracellular space.

## P2ry1 mediates coordinated neuronal activation and precise burst firing of SGNs

To evaluate the role of P2RY1 in initiating coordinated cellular activity in the cochlea, we monitored large-scale activity patterns in excised cochleae from *Pax2-Cre;R26-lsl-GCaMP3* mice, which express GCaMP3 in nearly all cells of the inner ear. Time lapse imaging revealed that the spontaneous $Ca^{2+}$ elevations that occur simultaneously within groups of ISCs, IHCs, and SGNs (*Eckrich et al., 2018*; *Tritsch and Bergles, 2010*; *Zhang-Hooks et al., 2016*) (*Figure 5A*) were abolished following inhibition of P2RY1 with MRS2500 (*Figure 5B,C*) and were dramatically reduced in *P2ry1* KO mice (*Pax2-Cre;R26-lsl-GCaMP3;P2ry1⁻/⁻*) (*Figure 5—figure supplement 1A,B*). Moreover, in accordance with the progressive increase in extracellular $K^+$ that follows P2RY1 inhibition, there was a gradual increase in spontaneous, uncoordinated $Ca^{2+}$ transients in IHCs in the presence of MRS2500 (*Figure 5D–F*), suggesting that this $K^+$ accumulation increases IHC firing. Similarly, IHCs in *P2ry1* KO mice displayed a higher level of uncorrelated $Ca^{2+}$ transients in hair cells (*Figure 5—figure supplement 1C–E*), indicative of enhanced excitability. Together, these results indicate that P2RY1 is required for coordinated activation of ISCs, IHCs, and SGNs before hearing onset and that P2RY1 inhibition leads to higher rates of uncorrelated activity.

IHCs in the developing cochlea exhibit regenerative $Ca^{2+}$ spikes that strongly activate post-synaptic SGNs, resulting in bursts of action potentials that propagate to the CNS. To determine if P2RY1 initiates burst firing in SGNs, we recorded spontaneous activity from SGNs using juxtacellular recordings from their somata (*Figure 6A*). Application of MRS2500 resulted in a dramatic reduction of high frequency burst firing in SGNs, visible as a decrease in burst frequency and action potentials per burst (*Figure 6E,F*). All SGN spiking was abolished by the AMPA receptor antagonist NBQX (50 µM) (*Figure 6D*), indicating that their activity requires synaptic excitation by IHCs. The precise patterning of action potentials within bursts was also disrupted by P2RY1 inhibition, as there were fewer

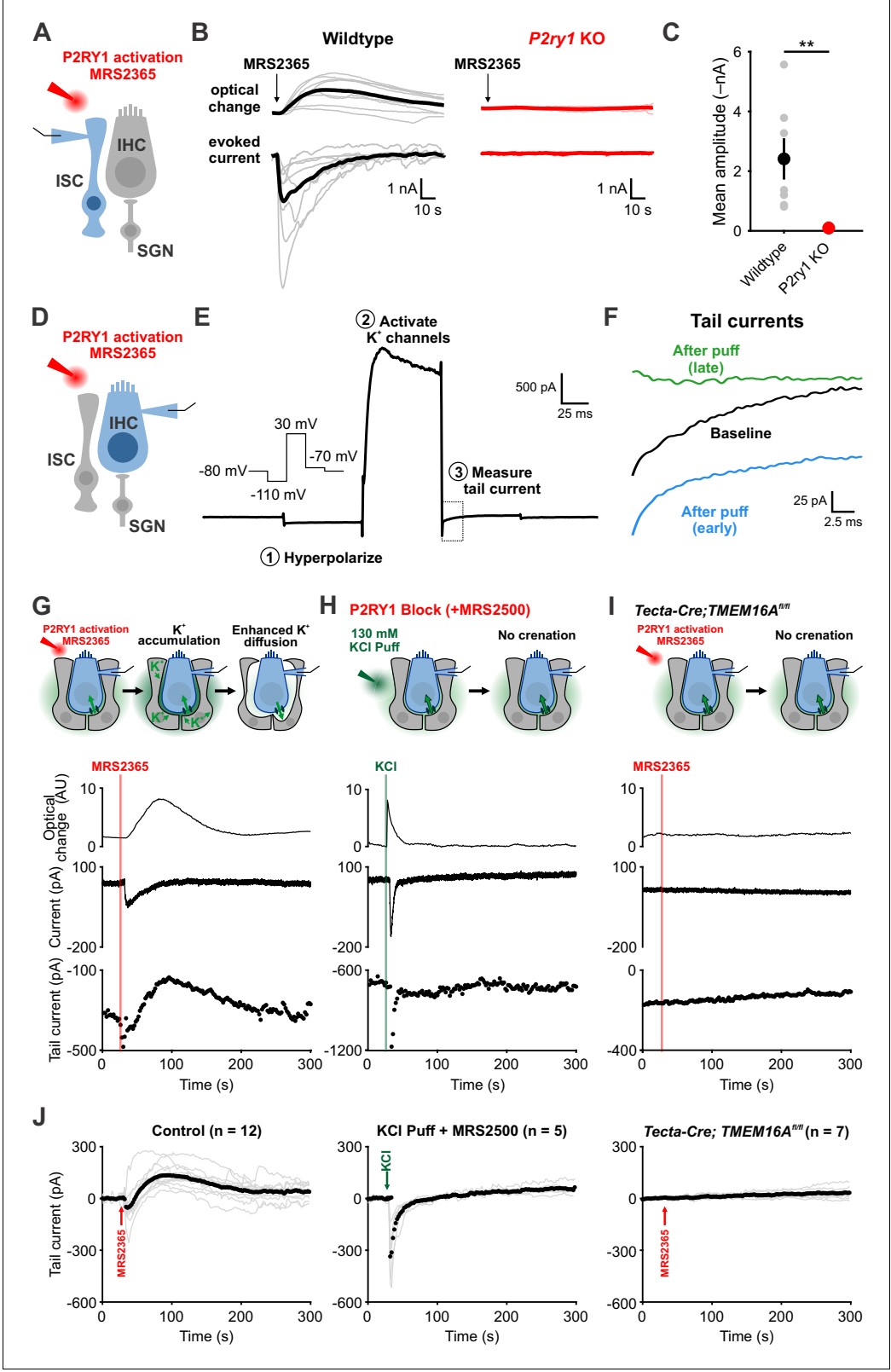

**Figure 4.** Activation of P2RY1 results in an initial accumulation of extracellular $K^+$, followed by crenation and enhanced $K^+$ clearance. (**A**) Schematic of whole-cell recording configuration from ISCs with puffs of MRS2365 (10 μM), a P2RY1 agonist. (**B**) Optical change (crenation) and current elicited with MRS2365 puffs in wildtype and *P2ry1* KO mice (**C**) Plot of mean current amplitude with MRS2365 puffs. n = 8 ISCs, 8 cochleae from 4 P6-8 wildtype mice

*Figure 4 continued on next page*

*Figure 4 continued*

and n = 7 ISCs, 7 cochleae from four from P2ry1 KO mice (two-tailed Student's t test; **p<0.005). (D) Schematic of whole-cell recording configuration from IHCs with puffs of MRS2365 (10 µM). (E) Example current trace and voltage-protocol designed to measure potassium accumulation. This protocol consisted of: (1) a hyperpolarizing step to −110 mV to relieve $K^+$ channel inactivation, (2) a depolarizing step to +30 mV to activate outward $K^+$ currents, and (3) a step to −70 mV to obtain a 'tail' current. Dashed box indicated tail current measurement period indicated in (F). (F) Tail currents observed during baseline, immediately following the MRS2365 puff (with 2 s), and after the puff (30 s). (G) Model of $K^+$ dynamics following MRS2365 stimulation. Initially, extracellular $K^+$ rapidly increases following stimulation, but ISCs crenate, increasing the amount of extracellular space and $K^+$ buffering. (bottom) Exemplar optical change (crenation), holding current, and tail current as a function of time with respect to MRS2365 puff. (H) Similar to G, but with KCl puffs (130 µM) in cochleae treated with MRS2500. (I) Similar to G, but in *Tecta-Cre;TMEM16A$^{fl/fl}$* mice where TMEM16A has been conditionally removed from the sensory epithelium (see *Figure 4—figure supplement 1*). No crenations were observed with MRS2365 stimulation. (J) Group tail currents across conditions from G-I. Gray lines indicate individual recordings; black points indicate the mean. Baseline was normalized to 0 pA for all traces. n = 12 IHCs, 11 cochleae from 9 P6-8 wildtype mice with MRS2365 stimulation, n = 5 IHCs, 5 cochleae from five wildtype mice with KCl stimulation, and n = 8 IHCs, 8 cochleae from 5 *Tecta-Cre;TMEM16A$^{fl/fl}$* mice with MRS2365 stimulation. See *Figure 4—source data 1* for plotted values and statistics.

The online version of this article includes the following source data and figure supplement(s) for figure 4:

**Source data 1.** Plotted values and statistics for *Figure 4*.

**Figure supplement 1.** Crispr-Cas9 mediated generation of the supporting cell specific Tecta-Cre mouse line.

---

interspike intervals in the 75–125 ms range (*Figure 6C,F*), which correspond to the maximum rate of $Ca^{2+}$ spike generation by IHCs during ATP-mediated excitation (*Tritsch et al., 2010b*). Additionally, the coefficient of variation measured for interspike intervals was significantly lower following P2RY1 inhibition, suggesting SGNs fire more randomly (*Figure 6E*). However, the average frequency of action potentials remained unchanged during P2RY1 inhibition (*Figure 6E*) due to increases in non-burst firing, consistent with depolarizing inward currents observed in IHCs with P2RY1 inhibition (*Figure 3F,K*). SGNs in *P2ry1* KO cochleae exhibited activity similar to wildtype SGNs in the presence of MRS2500, with a lower burst firing rate, fewer interspike intervals in the 75–125 ms range, and a lower coefficient of variation of interspike intervals relative to controls (*Figure 6G–I*). However, despite the profound contribution of P2RY1 to ISC and IHC activity, some burst-like behavior (clustered, but lacking the same density of action potentials in control conditions) was still observed in SGNs (*Figure 6C,D,G*) suggesting that other forms of excitation emerge in the absence of P2RY1, perhaps due to the increase in overall excitability or compensatory developmental changes. Together, these data indicate that P2RY1 is required to generate discrete bursts of action potentials in SGNs and that loss of these receptors enhances uncorrelated firing.

## P2RY1 promotes auditory neuron firing in vivo

The highly synchronized electrical activity exhibited by IHCs prior to hearing onset propagates through the entire developing auditory system to induce correlated firing of auditory neurons within isofrequency zones (*Babola et al., 2018*; *Tritsch et al., 2010b*). To determine if P2RY1 is required to produce this form of correlated activity, we used in vivo wide-field epifluorescence microscopy of the inferior colliculus (IC) in mice that express GCaMP6s in all neurons (*Snap25-T2A-GCaMP6s* and *Snap25-T2A-GCaMP6s;P2ry1$^{−/−}$* mice). Time lapse imaging revealed that both control and *P2ry1* KO mice exhibited correlated neuronal activity confined to stationary bands oriented along the tono-topic axis (*Figure 7A–C*). Spontaneous events were less frequent in *P2ry1* KO mice (9.7 ± 0.8 events per minute compared to 13.4 ± 0.7 events per minute in control; two-tailed Student's t test, p=0.002), although the events were similar in amplitude and duration (half-width) (*Figure 7D*), sug-gesting that some compensatory amplification of events occurs in these mice. Spontaneous activity in *P2ry1* KO mice differed from controls in three other ways. First, the contralateral bias exhibited for each event was higher, with the weaker relative to stronger side amplitude decreasing from 0.61 ± 0.02 to 0.44 ± 0.02 (two-tailed Student's t test, p=3.0e-6) (*Figure 7D*). Second, the coefficient of variation (ratio of standard deviation to the mean) of event amplitudes was 40% higher relative to controls (*Figure 7D*). Third, a detailed examination of the spatial location of events across the

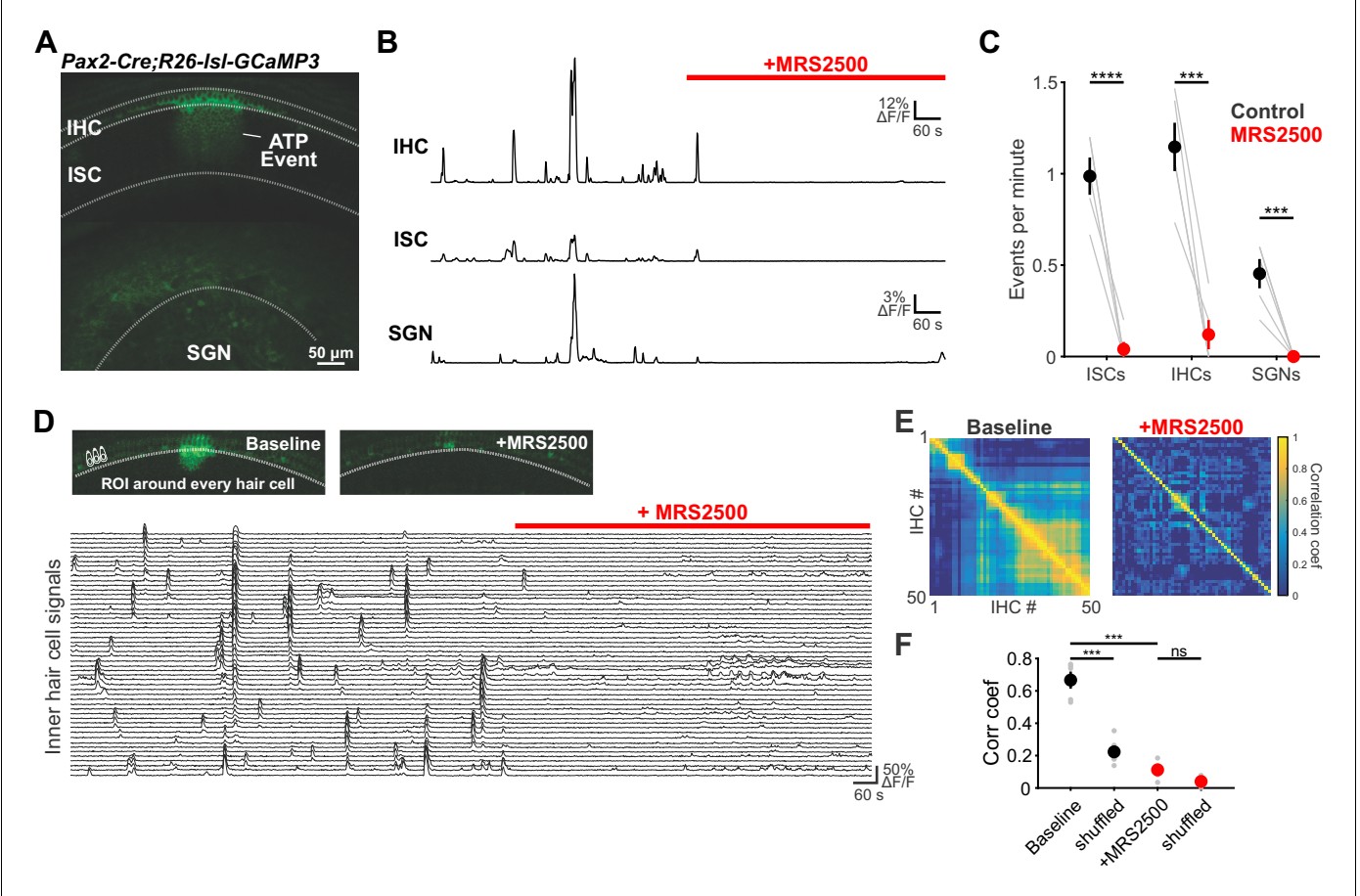

**Figure 5.** Large-scale coordinated activity in the cochlea requires P2RY1. (**A**) Exemplar $Ca^{2+}$ transient in excised cochlea from *Pax2-Cre;R26-lsl-GCaMP3* mice. Note the coordinated activation of ISCs, IHCs, and SGNs. (**B**) Traces of fluorescence intensity over time taken from ROIs that span the entire IHC, ISC, and SGN regions indicated in (**A**). (**C**) Plot of event frequency before and during application of MRS2500 (1 μM). n = 5 cochleae from 3 P5+1DIV (day in vitro) *Pax2-Cre;R26-lsl-GCaMP3* mice (two-tailed paired Student's t test with Bonferroni correction; ****p<5e-5, ***p<0.0005). (**D**) Exemplar images of IHC $Ca^{2+}$ transients. ROIs were drawn around every IHC for subsequent analysis (bottom). (**E**) Correlation matrices generated by calculating the linear correlation coefficient for all IHC pairs before and after MRS2500 application. (**F**) Plot of average correlation coefficient calculated between the four nearest IHCs. n = 5 P5+1DIV (day in vitro) cochleae from 3 *Pax2-Cre;R26-lsl-GCaMP3* mice (two-tailed paired Student's t test with Bonferroni correction; ***p<0.0005, ns, not significant). See *Figure 5—source data 1* for plotted values and statistics.

The online version of this article includes the following source data and figure supplement(s) for figure 5:

**Source data 1.** Plotted values and statistics for *Figure 5*.

**Figure supplement 1.** *P2ry1* KO mice exhibit reduced $Ca^{2+}$ transients in ISCs.

**Figure supplement 1—source data 1.** Plotted values and statistics for *Figure 5—figure supplement 1*.

tonotopic axis (*Figure 7E*) revealed that activity in brain areas later responsible for processing higher frequency tones (~8–16 kHz) exhibited the greatest reduction (68%) in *P2ry1 KO* mice, while activity in low frequency areas was unaltered (*Figure 7F–H*). In *P2ry1 KO* mice, bilateral removal of both cochleae abolished activity in the IC, demonstrating that activity in these mice still originates in the periphery (*Figure 7—figure supplement 1A–C*).

Although *P2ry1* KO mice mimic some aspects of acute P2RY1 inhibition, the absence of P2RY1 signaling throughout life may have led to compensatory changes, such as the increase in non-purinergic ISC activity (see *Figure 3—figure supplement 2E*). Therefore, to better assess the role of P2RY1 in generating spontaneous activity in vivo, we acutely inhibited these receptors by administering MRS2500 into the intraperitoneal cavity of mice while imaging activity in the IC. Compared to mice injected with control solution (5% mannitol), mice injected with MRS2500 exhibited dramatic reductions in IC event frequency (from 13.3 ± 0.8 to 3.9 ± 1.1 events per minute; two-tailed

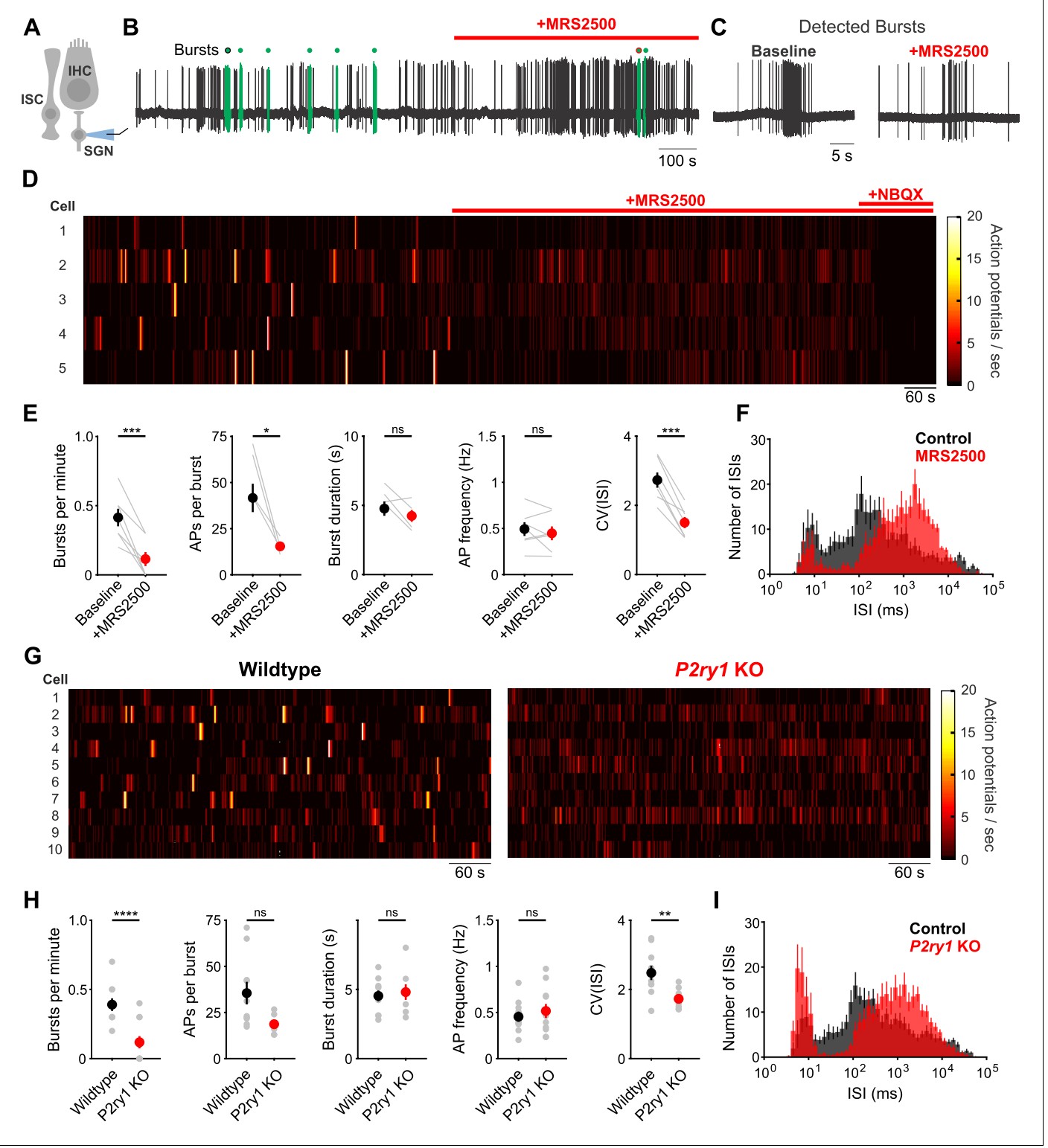

**Figure 6.** Inhibition of P2ry1 disrupts burst firing in spiral ganglion neurons. (**A**) Schematic of the juxtacellular recording configuration used to assess spiral ganglion neurons (SGNs). All SGN recordings were performed at room temperature. (**B**) Action potentials recorded before and during application of MRS2500 (1 μM). Detected bursts are indicated in green (see Materials and methods and Materials for parameters used for burst detection). Circle with black and red outlines are expanded in (**C**). (**C**) Action potentials within a detected burst before and after MRS2500 application. (**D**) Raster plots indicating the average firing rate of SGNs (bin: 1 s) before and during application of MRS2500 (1 μM) and subsequent NBQX (50 μM) (**E**) Plots of average burst frequency, burst duration, action potentials (AP) per burst, average AP frequency, and coefficient of variation for all interspike intervals

*Figure 6 continued on next page*

*Figure 6 continued*

(ISIs) measured. n = 7 SGNs, 7 P5+2DIV cochleae from six wildtype mice (two-tailed paired Student's t-test with Bonferroni correction; ***p<0.0005, *p<0.05, ns, not significant). (F) Average log-binned interspike interval histograms before and after MRS2500 application. (G) Raster plots indicating the average firing rate of SGNs (bin: 1 s) in wildtype and *P2ry1 KO* mice. (H) Plots of average burst frequency, burst duration, action potentials (AP) per burst, average AP frequency, and coefficient of variation for all ISIs measured. n = 10 SGNs, 10 P5+2DIV cochleae from wildtype mice and 11 SGNs, 10 cochleae from 9 *P2ry1* KO mice (two-tailed Student's t-test with Bonferroni correction; ****p<5e-5, **p<0.005, ns, not significant). (I) Average log-binned interspike interval histograms from wildtype and *P2ry1* KO SGN recordings. See *Figure 6—source data 1* for plotted values and statistics. The online version of this article includes the following source data for figure 6:

**Source data 1.** Plotted values and statistics for *Figure 6*.

Student's t test, p=0.0005) and amplitude (from 9.9 ± 0.5 to 4.9 ± 0.8% ΔF/F$_o$; two-tailed Student's t test, p=0.002)~5 min after administration (*Figure 8A–D*). This decrease was specific to the IC, as SC retinal wave activity (*Ackman et al., 2012*) was unaffected by acute MRS2500 administration (*Figure 8B,C,E*), suggesting that the locus of action is likely within the cochlea, which has been shown to have a permeable blood-tissue barrier at this age (*Suzuki et al., 1998*). Spatial analysis revealed that unlike the selective deficit observed in higher frequency zones in *P2ry1* KO mice, the inhibition with MRS2500 administration was not limited to certain tonotopic regions, but rather occurred evenly across all frequency zones (*Figure 8F,G*). Together, these data indicate that P2RY1 autoreceptors on ISCs within the cochlea play a critical role in initiating spontaneous bursts of neural activity in auditory centers within the brain prior to hearing onset.

## Discussion

Intense periods of neuronal activity dramatically alter the ionic composition of the extracellular environment, leaving behind excess K$^+$ that can alter neuronal excitability, induce spontaneous activity and trigger debilitating seizures. In the CNS, homeostatic control of extracellular K$^+$ levels is accomplished by glial cells, which redistribute K$^+$ passively through ion channels and actively through facilitated transport, but much less is known about the mechanisms that control excitability in the peripheral nervous system. Sensory hair cells and primary auditory neurons in the cochlea are surrounded by supporting cells that share key features with CNS glia and are thought to redistribute K$^+$ that accumulates during sound detection. However, prior to hearing onset, ATP-dependent K$^+$ release from these cells triggers periodic bursts of activity in nearby IHCs that propagate throughout the auditory system. Here, we demonstrate that this form of intrinsically generated activity is initiated through activation of P2RY1, a G$_q$-coupled metabotropic purinergic receptor. Acute inhibition or genetic removal of this receptor dramatically reduced spontaneous activity and disrupted burst firing in IHCs, SGNs and central auditory neurons. In addition to triggering episodic K$^+$-dependent depolarization of hair cells, activation of P2RY1 also enhanced K$^+$ clearance by increasing the volume of extracellular space, allowing more rapid dissipation of extracellular K$^+$ transients. This duality of purpose, to induce K$^+$ efflux and enhance K$^+$ clearance, promotes discrete bursts of activity throughout the developing auditory system.

### Purinergic signaling in the developing cochlea

Before the onset of hearing, neurons in the auditory system that will process similar sound frequencies exhibit periodic bursts of highly correlated activity, an entrainment that is initiated within the cochlea by the release of ATP (*Babola et al., 2018*; *Clause et al., 2014*; *Sonntag et al., 2009*; *Tritsch et al., 2010b*). The mechanism of ATP release from ISCs within the developing cochlear epithelium remains undefined, but may involve gap-junction hemichannels, as gap-junction antagonists profoundly inhibit spontaneous activity and lowering extracellular calcium, a manipulation that increases the open probability of hemichannels (*Peracchia, 2004*), markedly enhances the frequency of spontaneous activity (*Tritsch et al., 2007*). Extruded ATP then activates ISC purinergic receptors, triggering a rapid increase of intracellular Ca$^{2+}$, gating of TMEM16A Ca$^{2+}$-activated Cl$^-$ channels, and subsequent Cl$^-$ and K$^+$ efflux into the extracellular space (*Tritsch et al., 2007*; *Wang et al., 2015*). This transient K$^+$ efflux is sufficient to depolarize nearby IHCs, resulting in a burst of Ca$^{2+}$-action potentials, release of glutamate, and suprathreshold activation of postsynaptic SGNs via AMPA and NMDA receptors (*Tritsch et al., 2010b*; *Zhang-Hooks et al., 2016*) (*Figure 9A*). These

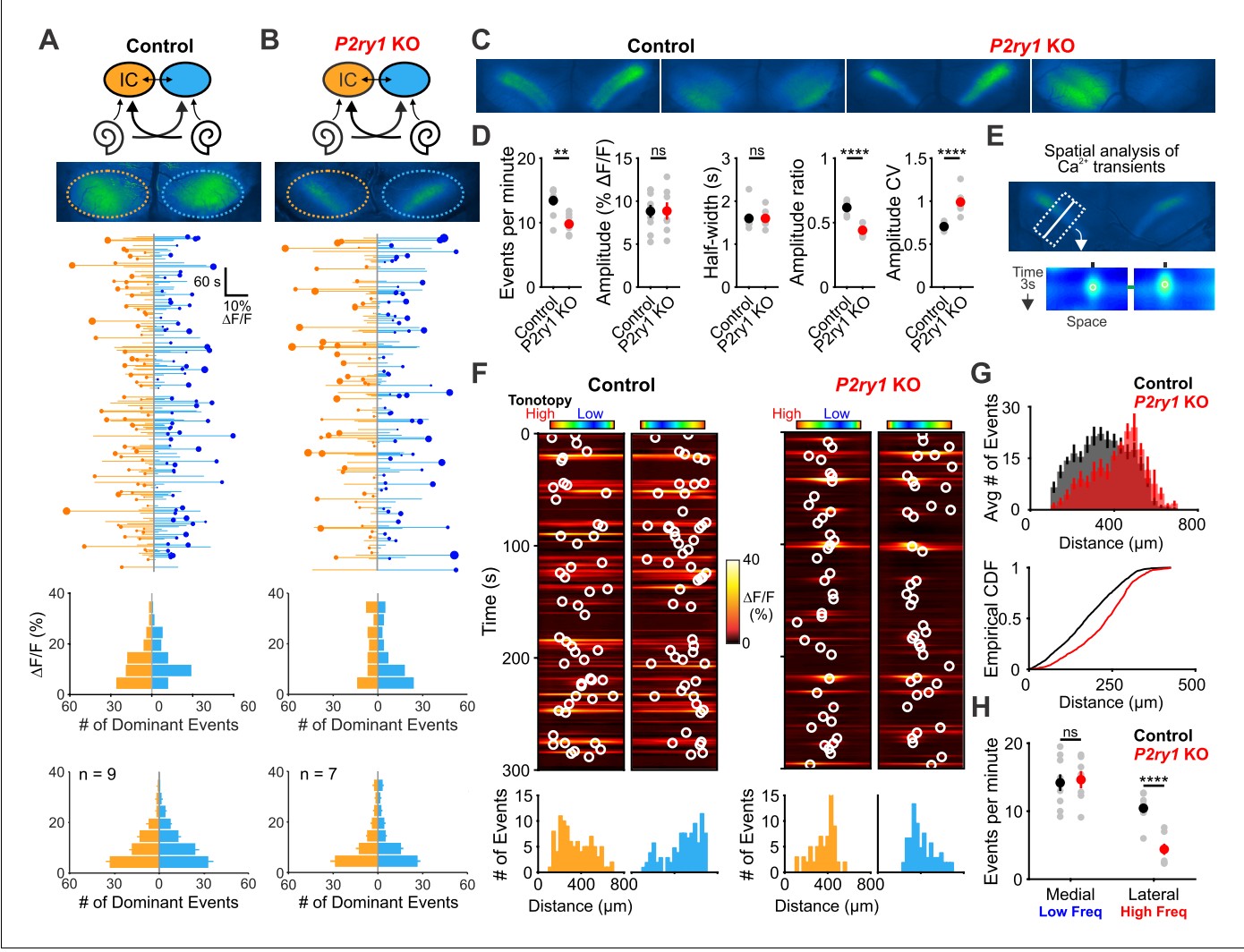

**Figure 7.** *P2ry1* KO mice exhibit reduced and spatially restricted spontaneous activity in the inferior colliculus. (**A**) Diagram illustrating flow of information through the auditory system and average intensity image over the 10 min imaging session. (middle) Activity over time in left and right IC in an individual where each line indicates the fluorescence intensity of each detected event; the circle indicates the dominant lobe, and the size of the circle indicated the difference in fluorescence. (bottom) Histograms showing the frequency of dominant events of a given amplitude for this experiment and for all experiments. Imaging was performed in *Snap25-T2A-GCaMP6s* mice (n = 9 mice). (**B**) Similar to (**A**), but in *Snap25-T2A-GCaMP6s;P2ry1⁻/⁻* (*P2ry1* KO) mice (n = 7 P6-8 mice). (**C**) Images of spontaneous events in the IC of in control (*Snap25-T2A-GCaMP6s*) and P2ry1 KO mice (*Snap25-T2A-GCaMP6s;P2ry1⁻/⁻*). (**D**) Comparisons of average frequency, amplitude, half-width, and event ratio from control and *P2ry1* KO mice. Bilateral amplitude ratio was calculated for events simultaneous across both lobes of the IC and defined as the ratio of the weak to the strong side amplitude. A ratio of 1 would indicate complete synchrony between lobes; a ratio of 0 would indicate complete asymmetry. n = 9 control and n = 7 *P2ry1* KO P6-8 mice (two-tailed Student's t test with Bonferroni correction; ****p<5e-4, **p<0.005, ns: not significant). (**E**) Exemplars of a single-banded event. Rectangular ROIs were placed as shown and averaged to create a 'line-scan' across the tonotopic axis. (bottom) Heat maps of activity as a function of time and distance; circles indicate detected peaks. (**F**) Activity over a five-minute time frame in the left and right IC of control and *P2ry1* KO mice. Circles indicate detected peaks. (bottom) Histograms of peak locations. (**G**) Histogram of average number of events across all control (black) and *P2ry1* KO (red) mice. (bottom) Cumulative distribution function of event locations across the tonotopic axis pooled from all animals. Events from left and right IC were combined for each experiment. (**H**) Quantification of event frequency in the medial (low frequency) and lateral (high frequency) regions of the IC. n = 9 control and 7 *P2ry1* KO P6-8 mice (two-tailed Student's t test with Bonferroni correction; ****p<5e-5, ns, not significant). See *Figure 7—source data 1* for plotted values and statistics.

The online version of this article includes the following source data and figure supplement(s) for figure 7:

**Source data 1.** Plotted values and statistics for *Figure 7*.

**Figure supplement 1.** Spontaneous activity in *P2ry1 KO* mice originates in the cochlea.

**Figure supplement 1—source data 1.** Plotted values and statistics for *Figure 7—figure supplement 1*.

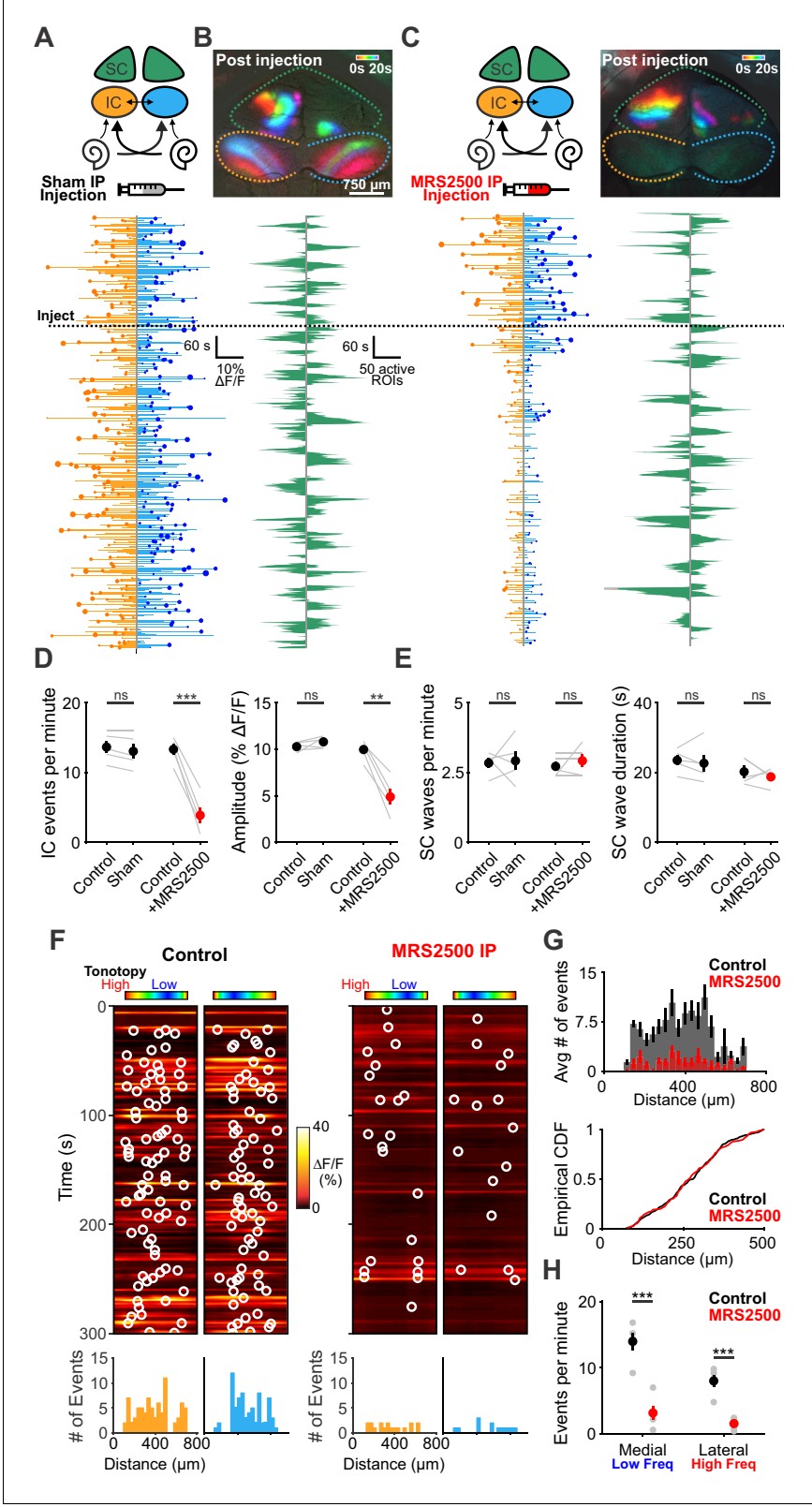

**Figure 8.** Delivery of MRS2500 in vivo dramatically reduces spontaneous activity in the developing auditory system. (**A**) Diagram illustrating flow of information to the midbrain and the visual superior colliculus. Sham solution (5% mannitol) was injected via IP catheter during imaging. (bottom) Activity over time in left and right IC in an individual where each line indicates the fluorescence intensity of each detected event; the circle indicates the

*Figure 8 continued on next page*

*Figure 8 continued*

dominant lobe, and the size of the circle indicated the difference in fluorescence. Dashed line indicates time of injection. (**B**) Calcium transients in the midbrain, color-coded based on time of occurrence following sham injection. (bottom) Calcium transients observed in the left and right SC. (**C**) Similar to (**A**) and (**B**), but with injection of MRS2500 (50 µL of 500 µM MRS2500 in 5% mannitol solution). (**D**) Plot of IC event frequency and amplitude in sham and MRS2500 injected animals. n = 5 mice for each condition (two-tailed paired Student's t test with Bonferroni correction; \*\*\*p<0.005, \*\*p<0.005, ns: not significant). (**E**) Plot of SC wave frequency and duration in sham and MRS2500 injected animals. n = 5 P6-8 mice for each condition (two-tailed paired Student's t test with Bonferroni correction; ns: not significant). (**F**) Activity along the tonotopic axis over a five-minute time frame in the left and right IC before (left) and after (right) MRS2500 injection. Circles indicate detected peaks. (bottom) Histograms of peak locations. (**G**) Histogram of average number of events before (black) and after (red) MRS2500 injection. (bottom) Cumulative distribution function of event locations across the tonotopic axis pooled from all animals. Events from left and right IC were combined for each experiment. (**H**) Quantification of event frequency in the medial (low frequency) and lateral (high frequency) regions of the IC. n = 5 P6-8 mice (two-tailed Student's t test with Bonferroni correction; \*\*\*p<0.005). See *Figure 8—source data 1* for plotted values and statistics.

The online version of this article includes the following source data for figure 8:

**Source data 1.** Plotted values and statistics for *Figure 8*.

bursts propagate throughout the entire developing auditory system (*Babola et al., 2018*) and likely activate the olivocochlear efferent reflex arc, providing bursts of inhibition to IHCs that may modulate the envelope of excitation (*Clause et al., 2014*) and/or the number of IHCs activated during each event. Our results show that activation of metabotropic P2RY1 autoreceptors is a key first step in this transduction pathway. P2RY1 is highly expressed by ISCs at a time when spontaneous activity is prominent in the cochlea (*Scheffer et al., 2015*; *Tritsch and Bergles, 2010*) (*Figure 2A*), and spontaneous activity was reduced when intracellular $Ca^{2+}$ stores were depleted or PLC was inhibited (*Figure 1B–F*), manipulations that disrupt canonical $G_q$-coupled GPCR signaling pathways (*Erb and Weisman, 2012*; *Fabre et al., 1999*). Moreover, our pharmacological studies indicate that P2RY1 is both necessary and sufficient for spontaneous current generation in supporting cells (*Figures 3B* and *4B*), and acute inhibition of P2RY1 in vivo profoundly decreased cochlea-generated activity in the auditory midbrain (*Figure 8C*). This reliance on P2RY1 is somewhat unexpected, as ionotropic P2X and other metabotroic P2Y receptors are also widely expressed in the developing cochlea (*Brändle et al., 1999*; *Eckrich et al., 2018*; *Huang et al., 2010*; *Lahne and Gale, 2008*; *Liu et al., 2015*; *Nikolic et al., 2003*; *Scheffer et al., 2015*; *Tritsch et al., 2007*). The lack of P2X or other $G_q$-coupled P2Y receptor engagement may reflect the particular spatial-temporal characteristics of ATP

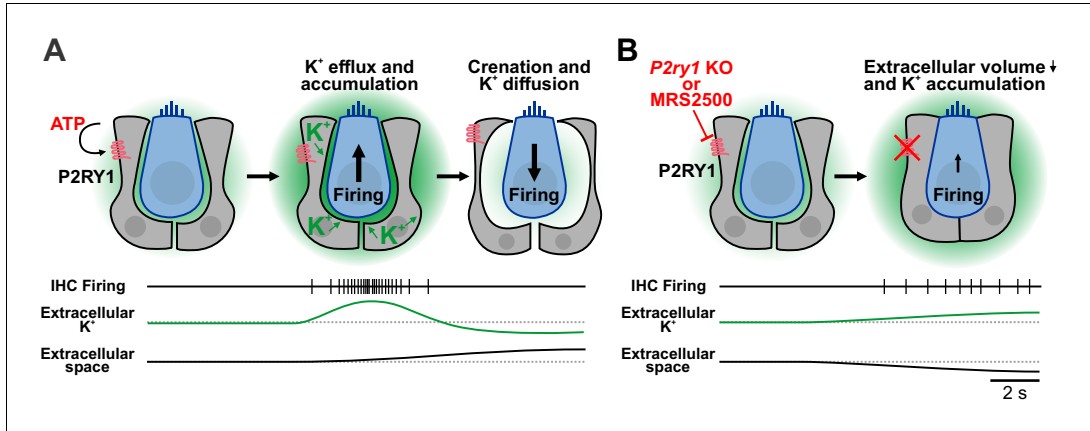

**Figure 9.** Summary of the dualistic action of P2Y1 signaling. (**A**) Spontaneous release of ATP and P2RY1 activation of ISCs initiates a cascade of intracellular signaling events that culminates in a transient increase in extracellular $K^+$ levels and depolarization of IHCs, resulting in burst firing. The large efflux of ions during each ATP event draws water out of ISCs through osmosis, leading to pronounced shrinkage (crenation) of ISCs and enhanced diffusion of $K^+$ through the increased extracellular volume. (**B**) Inhibition or genetic removal of P2RY1 induces cellular swelling and loss of extracellular space, leading to extracellular $K^+$ accumulation and tonic firing of IHCs.

release by ISCs, which may occur in locations enriched in P2RY1 or yield ATP concentration transients that favor P2RY1 activation. Exogenous ATP can induce all of the phenomenon associated with spontaneous events (ISC currents, crenation, IHC depolarization, SGN burst firing); however, it is possible that other nucleotides are released that have greater affinity for P2RY1 (e.g. ADP), or that extracellular nucleotidases rapidly convert ATP to ADP that favor activation of native metabotropic receptors (*von Kügelgen, 2006*; *Vlajkovic et al., 1998*; *Vlajkovic et al., 2002*).

## Control of extracellular $K^+$ dynamics by supporting cells

Pharmacological inhibition of P2RY1 unexpectedly induced IHCs to gradually depolarize and begin tonic, uncorrelated firing, a phenotype also observed in *P2ry1* KO mice (*Figure 5—figure supplement 1C*). Our studies indicate that this phenomenon occurs because P2RY1 controls the volume of the extracellular space in the organ of Corti. Activation of P2RY1 induces ISCs to shrink osmotically (crenate; *Figure 9A*), a consequence of ion and water efflux that is triggered by opening of TMEM16A channels (*Figure 4G*). The resulting increase in extracellular space lasts for many seconds and enhances dissipation of extracellular $K^+$ (*Figure 9A*), visible through the time-dependent shift in the reversal potential of $K^+$-mediated tail currents (*Figure 4G,I*). Conversely, inhibition of P2RY1 increased the size of ISCs, a swelling-induced 'relaxation' that concomitantly decreased extracellular space around IHCs (*Figure 9B*). $K^+$ accumulation and depolarization of IHCs followed, an effect absent when IHC $K^+$ channels were inhibited (*Figure 3J,K*) or $K^+$ release from cells in the organ of Corti was reduced (*Figure 3H–K*), leading to tonic firing of IHCs and post-synaptic SGNs (*Figure 6D,G* and *Figure 9B*). This phenomenon is consistent with the depolarizing shift in resting membrane potential of IHCs observed in *Tmem16A* cKO mice (*Wang et al., 2015*), which similarly blocks ISC crenation. Of note, epileptiform activity can be elicited in the CNS by inducing cell swelling with hypoosmotic solutions or by impairing $K^+$ buffering (*Larson et al., 2018*; *Murphy et al., 2017*; *Thrane et al., 2013*). Basal P2RY1 activation in supporting cells hyperpolarizes nearby IHCs by expanding the extracellular space and lowering local $K^+$ concentrations. These changes increase the dynamic range of IHCs, allowing them to respond to a wider range of ATP release events and enabling finer control of their excitability through transient ATP-mediated signaling events.

The tonic inward current that develops in ISCs in response to P2RY1 block was abolished when homeostatic $K^+$ release pathways ($Na^+$ channels, $Ca^{2+}$ channels, $Na^+$-$K^+$-$Cl^-$ cotransporters, and Na, K-ATPase) were inhibited (*Figure 3H,I*), suggesting that $K^+$ redistribution mechanisms, in the absence of ISC crenation, are weak at this stage of development. Indeed, although the membrane potential of ISCs is close to $E_K$, their membrane conductance is dominated by intercellular gap junction channels; when uncoupled from their neighbors, they exhibit very high (1–2 GΩ) input resistance (*Jagger and Forge, 2015*; *Wang et al., 2015*), suggesting that few $K^+$ leak channels are expressed. The presence of tight junctions at the apical surface of the cochlear epithelium and the limited $K^+$ conductance of ISCs may restrict passive diffusion and dilution of $K^+$, similar to what has been described in the vestibular epithelium (*Contini et al., 2017*), thus necessitating uptake via alternative mechanisms. Both inner phalangeal and Deiters' cells (which envelop the inner and outer hair cells, respectively) express $K^+$-$Cl^-$ symporters, Na,K-ATPase pumps, and inwardly-rectifying $K^+$ channels that may siphon $K^+$ into the supporting cell syncytium after extrusion from hair cells. However, the apparently low capacity of these systems places a greater dependence on diffusion within the extracellular volume fraction controlled by the supporting cells.

Similar to ISCs, astrocytes in the CNS facilitate rapid dissipation of extracellular $K^+$ by $K^+$ uptake and redistribution through the glial syncytium via gap junctions, a mechanism termed spatial buffering (*Kofuji and Newman, 2004*). Astrocytes are efficient $K^+$ sinks due to their highly negative resting potential (∼—85 mV) and large resting $K^+$ conductance dominated by inward rectifying $K^+$ channels (*Olsen, 2012*) and two-pore leak $K^+$ channels (*Ryoo and Park, 2016*). While uptake of $K^+$ through these channels is passive, recent studies suggest that $K^+$ buffering in astrocytes is actively regulated by purinergic receptors. Following stimulation of native astrocyte purinergic receptors or foreign $G_q$-coupled receptors (MrgA1) and release of $Ca^{2+}$ from intracellular stores, Na,K-ATPase activity increased, resulting in a transient decrease in extracellular $K^+$, hyperpolarization of nearby neurons, and reduction in their spontaneous activity (*Wang et al., 2012*). Although P2RY1 is expressed by some astrocytes and can trigger $Ca^{2+}$ waves (*Gallagher and Salter, 2003*), this mechanism does not appear to regulate IHC excitability in the cochlea, as stimulation of P2RY1 in *Tmem16a* cKO mice, which have intact metabotropic receptor signaling but no crenations (*Wang et al., 2015*), did not

hyperpolarize IHCs (*Figure 4I,J*). Thus, astrocytes and cochlear ISCs use purinergic signaling in distinct ways to maintain the ionic stability of the extracellular environment and control the excitability of nearby cells.

## Role of supporting cells in the generation of spontaneous activity

Our understanding of how non-sensory cells contribute to spontaneous activity has been limited by a lack of in vivo mechanistic studies. Recent advances in visualizing cochlea-induced spontaneous activity in central auditory centers in vivo using genetically-encoded calcium indicators (*Babola et al., 2018*) allowed us to assess whether supporting cell purinergic receptors are involved in generating this activity. Prior to hearing onset, the blood-labyrinth barrier within the inner ear is not fully formed (*Suzuki et al., 1998*), permitting pharmacological access to the cochlea at this age. Infusion of a P2RY1 inhibitor into the intraperitoneal space dramatically decreased the frequency and amplitude of transient, spatially restricted bands of activity in the inferior colliculus within minutes, while retina-induced activity in the superior colliculus (*Ackman et al., 2012*) was unaffected (*Figure 8*), suggesting that inhibition is not due to activation of astrocyte P2RY1 receptors or broad disruption of neural activity; as noted above, inhibition of P2RY1 in astrocytes would be expected to enhance, rather than inhibit neuronal activity (*Wang et al., 2012*). Within the CNS, *P2ry1* mRNA is expressed in the cochlear nucleus and P2RY1-mediated calcium transients can be induced in spherical bushy cells (*Milenkovic et al., 2009*). Although the consequences of P2RY1 inhibition on neuronal activity in auditory centers is not yet known, it is possible that some effects of MRS2500 also result from actions on P2RY1 in the CNS.

In vivo imaging in *P2ry1* KO mice recapitulated many aspects of changes seen when P2RY1 was acutely inhibited, with significantly reduced neuronal activity observed in lateral regions of the IC (later active to 8–16 kHz tones; *Figure 7*). However, neuronal burst firing persisted within central regions of the IC, regions that will ultimately process lower frequency sounds (3–8 kHz). At present, we do not know the reason for the differences between P2RY1 inhibition and constitutive *P2ry1* deletion, but the absence of P2RY1 throughout development may have induced widespread compensatory changes as a result of the new patterns of activity experienced by these developing neurons (*Figure 3—figure supplement 2*). Indeed, developing sensory systems exhibit a remarkable ability to preserve spontaneous activity. In the visual system, cholinergic antagonists injected directly into the eye blocks retinal waves in vivo (*Ackman et al., 2012*), but genetic removal of the β2 acetylcholine receptor subunit alters, but does not abolish, peripherally-generated activity (*Zhang et al., 2012*). In the auditory system, in vivo spontaneous activity can be blocked by acute inhibition of cochlear AMPARs, but deaf mice in which hair cell glutamate release is abolished (*Vglut3* KO mice) exhibit activity patterns remarkably similar to control mice (*Babola et al., 2018*). These robust homeostatic mechanisms allow spontaneous activity to persist despite disruption of key transduction components. Local purinergic signaling within the cochlea may still initiate tonotopic activity in central auditory circuits of *P2ry1* KO mice, perhaps by engaging other subtypes of metabotropic purinergic receptors, as events in the IC exhibited spatial and temporal characteristics similar to controls. IHCs and SGNs are more depolarized in these mice, reducing the threshold for activation by other purinergic receptors. The role of P2RY1 in the developing CNS remains unexplored, but potential compensatory effects may be compounded in central circuits, due to loss of P2RY1 in astrocytes and neurons. Although such gain-of-function changes in the developing nervous system present challenges for interpretation of genetic manipulations, the preservation of early, patterned activity in children that carry deafness mutations may improve the outcome of later therapeutic interventions to restore hearing.

## Purinergic receptors in the adult cochlea

In the adult inner ear, members of all purinergic receptors subtypes (ionotropic P2X receptors, metabotropic P2Y, and adenosine P1 receptors) are expressed by cells throughout the sensory epithelium, Reisner's membrane, stria vascularis, and SGNs (*Housley et al., 2009*; *Huang et al., 2010*). The widespread expression of these receptors coupled with observations of increased endolymphatic ATP concentrations following trauma (*Muñoz et al., 1995*) have led to the hypothesis that these receptors serve a neuroprotective role. Indeed, infusion of ATP into the inner ear profoundly reduces sound-evoked compound action potentials in the auditory nerve (*Bobbin and Thompson,*

*1978*; *Muñoz et al., 1995*), presumably due to decreased endolymphatic potential following shunting inhibition through P2RX2 (*Housley et al., 2013*) or excitotoxic damage (*Cisneros-Mejorado et al., 2015*) to IHCs or SGNs. Consistent with these observations, P2RX2 null mice and humans with a P2RX2 variant (c.178G > T) experience progressive sensorineural hearing loss (*Yan et al., 2013*). $Ca^{2+}$ imaging and recordings from adult cochleae have also revealed robust responses to UTP in the inner sulcus, pillar cells, and Deiters' cells (*Sirko et al., 2019*; *Zhu and Zhao, 2010*), suggesting that metabotropic purinergic receptors continue to be expressed. Following traumatic noise damage, ATP release could activate $K^+$ buffering mechanisms in supporting cells, enhance $K^+$ redistribution, reduce IHC depolarization and prevent excitotoxic damage. Purinergic receptors may also contribute to IHC gain control by influencing their membrane potential, as ATP circulates in the endolypmph at low nanomolar concentrations (*Muñoz et al., 1995*). Further studies involving conditional deletion of *P2ry1* from ISCs in the adult cochlea may help to define the role of this receptor in both normal hearing and injury contexts.

# Materials and methods

## Key resources table

| Reagent type (species) or resource | Designation | Source or reference | Identifiers | Additional information |
|---|---|---|---|---|
| Strain, strain background (Rattus norvegicus) | Sprague Dawley | Charles River | RRID:MGI:5651135 | |
| Strain, strain background (Mus musculus) | FVB mice | Charles River | RRID:IMSR_CRL:207 | |
| Genetic reagent (Mus musculus) | P2ry1tm1Bh;P2ry1 KO | (*Fabre et al., 1999*) | RRID:MGI:3623279 | |
| Genetic reagent (Mus musculus) | Pax2-Cre | (*Ohyama and Groves, 2004*) | RRID:IMSR_RBRC09434 | |
| Genetic reagent (Mus musculus) | R26-lsl-GCaMP3 | (*Paukert et al., 2014*) | | |
| Genetic reagent (Mus musculus) | B6.Cg-Snap25tm3.1Hze/J; Snap25-T2A-GCaMP6s | Jackson Laboratory | RRID:IMSR_JAX:025111 | |
| Genetic reagent (Mus musculus) | TMEM16Afl/fl | (*Schreiber et al., 2015*) | | |
| Genetic reagent (Mus musculus) | Tecta-Cre | this paper | | A mouseline with Cre expression limited primarily to the sensory epithelium. Cross to TdTomato line reveals sparse labeling of cells within the temporal bone and spiral ganglion neurons. |
| Genetic reagent (Mus musculus) | P2ry1tm1(KOMP) Vlcg;P2ry1 LacZ | KOMP | RRID:IMSR_KOMP: VG12793-1-Vlcg | |
| Genetic reagent (Mus musculus) | B6.Cg-Gt(ROSA) 26Sortm14 (CAG-tdTomato)Hze/J; Ai14; TdTomato | Jackson Laboratory | RRID:IMSR_JAX:007909 | |
| Antibody | Chicken polyclonal anti-beta-gal | Aves | Cat:BGL-1010; RRID:AB_2313508 | (1:4000) |
| Antibody | Rabbit polyclonal anti-Myosin-VIIa | Proteus Biosciences | Cat:25–6790; RRID:AB_2314838 | (1:500) |
| Antibody | Donkey anti-chicken Alexa Fluor 488 | Life Technologies | Cat:ab150073; RRID:AB_2636877 | (1:2000) |
| Antibody | Donkey anit-rabbit Alexa Fluor 546 | Jackson ImmunoResearch | Cat:711-165-152; RRID:AB_2307443 | (1:2000) |

*Continued on next page*

*Continued*

| Reagent type (species) or resource | Designation | Source or reference | Identifiers | Additional information |
|---|---|---|---|---|
| Sequence-based reagent | Primer: cccagttgagattggaaagtg (Snap25GC6s-com-s) | Jackson Laboratory | | |
| Sequence-based reagent | Primer: acttcgcacaggatccaaga (Snap25GC6s-mut-as) | Jackson Laboratory | | |
| Sequence-based reagent | Primer: ctggttttgttggaatcagc (Snap25GC6s-wt-as) | Jackson Laboratory | | |
| Sequence-based reagent | Primer: ccgtcaggacaattatcacc (P2ry1-com-as) | this paper | | |
| Sequence-based reagent | Primer: cctaccagccctcatcttct (P2ry1-wt-s) | this paper | | |
| Sequence-based reagent | Primer: cttctatcgccttcttgacg (P2ry1-KO-s) | this paper | | |
| Sequence-based reagent | Primer: gatggttgtggtgtgtctcg (Tecta-com-s) | this paper | | |
| Sequence-based reagent | Primer: cagtgatgagggaggaggtg (Tecta-wt-as) | this paper | | |
| Sequence-based reagent | Primer: cctgtccctgaacatgtcca (Tecta-Cre-as) | this paper | | |
| Sequence-based reagent | Primer: gctgcctgagttggaaagaa (P2ry1-LacZ-com-s) | this paper | | |
| Sequence-based reagent | Primer: ggcttcatgtggaaaacgaa (P2ry1-LacZ-wt-as) | this paper | | |
| Sequence-based reagent | Primer: ctctgctgcctcctggcttct (Rosa26-s) | *Paukert et al., 2014* | | |
| Sequence-based reagent | Primer: cgaggcggatcacaagcaata (Rosa26-as) | *Paukert et al., 2014* | | |
| Sequence-based reagent | Primer: tcaatgggcgggggtcgtt (CMV-E-as) | *Paukert et al., 2014* | | |
| Sequence-based reagent | Primer: attcagacggcaaacgactg (P2ry1-LacZ-LacZ-as) | this paper | | |
| Sequence-based reagent | crRNA: TAATGATGAATAATTCATCC (Tecta exon two targeting) | this paper | | |
| Chemical compound, drug | BAPTA-AM | Sigma | Cat:A1076; CAS:126150-97-8 | (100 µM) |
| Chemical compound, drug | Thapsigargin | Sigma | Cat:T9033; CAS:67526-95-8 | (2 µM) |
| Chemical compound, drug | U73122 | Tocris | Cat:1268; CAS:112648-68-7 | (10 µM) |
| Chemical compound, drug | U73343 | Tocris | Cat:4133; CAS:142878-12-4 | (10 µM) |
| Chemical compound, drug | Ryanodine | Tocris | Cat: 1329; CAS:15662-33-6 | (10 µM) |
| Chemical compound, drug | MRS2500 | Tocris | Cat:2159; CAS:630103-23-0 | (1 µM) |
| Chemical compound, drug | PPADS | Sigma | Cat:P178; CAS:192575-19-2 (anhydrous) | (100 µM) |
| Chemical compound, drug | Suramin | Sigma | Cat:S2671; CAS:129-46-4 | (50 µM) |
| Chemical compound, drug | TTX | Abcam | Cat: ab120055; CAS: 18660-81-6 | (1 µM) |

*Continued on next page*

*Continued*

| Reagent type (species) or resource | Designation | Source or reference | Identifiers | Additional information |
|---|---|---|---|---|
| Chemical compound, drug | Ouabain | Tocris | Cat: 1076; CAS: 630-60-4 | (10 μM) |
| Chemical compound, drug | Bumetanide | Tocris | Cat: 3108; CAS: 28395-03-1 | (50 μM) |
| Chemical compound, drug | CsCl2 | Sigma | Cat: C4036; CAS: 7647-17-8 | (100 μM) |
| Software, algorithm | ZEN Blue/Black | Zeiss | RRID:SCR_013672 | |
| Software, algorithm | ImageJ | https://imagej.nih.gov/ij/ | RRID:SCR_003070 | |
| Software, algorithm | MultiStackReg | http://bradbusse.net/sciencedownloads.html | RRID:SCR_016098 | |
| Software, algorithm | MATLAB 2017b | Mathworks | RRID:SCR_001622 | |
| Software, algorithm | CorelDRAW Graphics Suite | Corel | RRID:SCR_014235 | |
| Strain, strain background (*Rattus norvegicus*) | Sprague Dawley | Charles River | RRID:MGI:5651135 | |

Both male and female mice and rats of postnatal days P6-P8 were used for all experiments and randomly allocated to experimental groups. All animals were healthy and were only used for experiments detailed in this study. Transgenic breeders were crossed to female FVB/NJ (Friend Virus B NIH Jackson; demonstrated low hearing thresholds at 28 weeks) mice to improve litter sizes and pup survival (*Zheng et al., 1999*). Mice were housed on a 12 hr light/dark cycle and were provided food ad libitum. This study was performed in accordance with the recommendations provided in the Guide for the Care and Use of Laboratory Animals of the National Institutes of Health. All experiments and procedures were approved by the Johns Hopkins Institutional Care and Use Committee (protocol #: M018M330). All surgery was performed under isoflurane anesthesia and every effort was made to minimize suffering.

## Electrophysiology

For inner supporting cell recordings, apical segments of the cochlea were acutely isolated from P6-P8 rat (*Figure 1*) and mouse pups (all other figures) and used within 2 hr of the dissection. Cochleae were moved into a recording chamber and continuously superfused with bicarbonate-buffered artificial cerebrospinal fluid (1.5–2 mL/min) consisting of the following (in mM): 119 NaCl, 2.5 KCl, 1.3 $MgCl_2$, 1.3 $CaCl_2$, 1 $NaH_2PO_4$, 26.2 $NaHCO_3$, 11 D-glucose and saturated with 95% $O_2$/5% $CO_2$ to maintain a pH of 7.4. Solutions were superfused at either room temperature or near physiological temperature (32–34°C) using a feedback-controlled in-line heater (Warner Instruments), as indicated in figure legends. Whole-cell recordings of inner supporting cells (ISCs) were made under visual control using differential interference contrast microscopy (DIC). Electrodes had tip resistances between 3.5–4.5 MΩ when filled with internal consisting of (in mM): 134 $KCH_3SO_3$, 20 HEPES, 10 EGTA, 1 $MgCl_2$, 0.2 Na-GTP, pH 7.3. Spontaneous currents were recorded with ISCs held at −80 mV.

For inner hair cell recordings, apical segments of the cochlea were acutely isolated from P6-P8 mouse pups and used within 2 hr of the dissection. Cochleae were moved into a recording chamber and continuously superfused with bicarbonate-buffered artificial cerebrospinal fluid (1.5–2 mL/min) consisting of the following (in mM): 115 NaCl, 6 KCl, 1.3 $MgCl_2$, 1.3 $CaCl_2$, 1 $NaH_2PO_4$, 26.2 $NaHCO_3$, 11 D-glucose. Solutions were saturated with 95% $O_2$/5% $CO_2$ to maintain a pH of 7.4. Solutions were superfused at room temperature. Electrodes had tip resistances between 4.5–6 MΩ when filled with internal consisting of (in mM): 134 $KCH_3SO_3$, 20 HEPES, 10 EGTA, 1 $MgCl_2$, 0.2 Na-GTP, pH 7.3. For hair cell recordings with $K^+$ channels inhibited with cesium and TEA, the internal solution consisted of (in mM): 100 cesium methanesulfonate, 20 TEA-Cl, 10 EGTA in CsOH, 20 HEPES, 1 $MgCl_2$, 0.2 Na-GTP, pH 7.3 with CsOH. Spontaneous currents were recorded with IHCs held at near their resting membrane potential (−75 to –80 mV).

Errors due to the voltage drop across the series resistance and the liquid junction potential were left uncompensated for recordings of spontaneous activity. For IHC recordings with K+ accumulation voltage protocols (*Figure 4*), the amplifier compensation circuit was used to compensate 70% of the access resistance. Recordings that displayed more than a 10% increase in access resistance or access resistances > 30 MΩ were discarded. ISC and IHC spontaneous currents were recorded with pClamp 10 software using a Multiclamp 700B amplifier, low pass filtered at 2 kHz, and digitized at 5 kHz with a Digidata 1322A analog-to-digital converter (Axon Instruments).

For SGN juxtacellular recordings, cochleae were dissected and cultured for 2 days (see Cochlear Explant Culture section below). Cochleae were then transferred to a recording chamber and continuously superfused with bicarbonate-buffered aCSF (same as ISCs) at 1.5–2 mL/min. Recordings were performed at room temperature. Electrodes for SGN recordings had tip resistances between 1.5–2.5 MΩ when filled with artificial cerebrospinal fluid. Extracellular potentials were recorded for 10 min with pClamp10 software using a Multiclamp 700B amplifier, low pass filtered at 20 kHz, and digitized at 50 kHz with a Digidata 1322A analog-to-digital converter (Axon Instruments). For MRS2500 experiments, spikes were analyzed in five-minute windows; five minutes of baseline preceded ten minutes of superfusion of aCSF containing 1 µM MRS2500. Firing behavior in the latter five minutes of MRS2500 was used for measurement.

Action potentials were analyzed offline using custom routines written in Matlab 2017b (Mathworks). Briefly, raw traces were high-pass filtered to remove baseline drift and spikes were identified using an amplitude threshold criterion. As described previously (*Tritsch and Bergles, 2010*), bursts were identified by classifying interspike intervals into non-bursting intervals (>1 s), burst intervals (30 ms - 1 s), and mini-burst intervals (<30 ms). Bursts were defined as clusters of at least 10 consecutive burst intervals (with mini-burst intervals being ignored in the context of burst detection). Spikes within mini-bursts were included when calculating the number of spikes within a burst. Colored raster plots were generated by grouping spikes into one-second bins and applying a color map to the resulting data (modified 'hot' colormap; Matlab).

## Cochlear explant culture

Cochleae were dissected from postnatal day 5–6 control (*P2ry1+/+* or *Pax2-Cre;R26-lsl-GCaMP3*) and *P2ry1* KO (*P2ry1−/−* or *Pax2-Cre;R26-lsl-GCaMP3;P2ry1−/−*) mice in ice-cold, sterile-filtered HEPES-buffered artificial cerebrospinal fluid (aCSF) consisting of the following (in mM): 130 NaCl, 2.5 KCl, 10 HEPES, 1 NaH$_2$PO$_4$, 1.3 MgCl$_2$, 2.5 CaCl$_2$, and 11 D-Glucose. Explants were mounted onto Cell-Tak (Corning) treated coverslips and incubated at 37° C for 24–48 hr in Dulbecco's modified Eagle's medium (F-12/DMEM; Invitrogen) supplemented with 1% fetal bovine serum (FBS) and 10 U/mL penicillin (Sigma) prior to recording or imaging.

## Transmitted light imaging

Cochlear segments were imaged with a Olympus 40x water immersion objective (LUMPlanFl/IR) and recorded using MATLAB and a USB capture card (EZ Cap). Difference movies were generated by subtracting frames at time $t_n$ and $t_{n+5}$ seconds using ImageJ software to generate an index of transmittance change over time. To quantify transmittance changes, a threshold of three standard deviations above the mean was applied to the values. To calculate the frequency of these events, the whole field was taken as an ROI and peaks were detected using MATLAB (findpeaks function). To calculate area of these events, a Gaussian filter (sigma = 2.0) was applied to the image after thresholding and the borders detected using MATLAB (bwlabel function). The area was then calculated as the number of pixels within the border multiplied by the area scaling factor (µm/pixel)$^2$ measured with a stage micrometer.

## Immunohistochemistry and X-gal reaction

Mice were deeply anesthetized with isoflurane and perfused with freshly prepared paraformaldehyde (4%) in 0.1 M phosphate buffer. Cochleae were post-fixed for 45 min at room temperature and stored at 4°C until processing. For X-gal reactions, P6-P8 cochleae were removed from the temporal bone and washed 3 × 5 min with PBS. Tissue was then incubated for 24 hr in the dark at 37°C in X-gal working solution consisting of (in mM): five potassium ferricyanide crystalline, five potassium ferricyanide trihydrate, two magnesium chloride, and 0.1% X-gal (GoldBio) dissolved in DMSO. After

washing 3 × 5 min with PBS, images of cochleae were acquired on a dissecting microscope (Zeiss Stemi 305). For immunohistochemistry, fixed tissue was washed 3 × 5 min in PBS, placed in 30% sucrose solution overnight, and incubated in OCT mounting medium overnight at 4°C. Ten micron thick cross-sections of the cochlea were made on a cryostat and mounted on Superfrost Plus slides (Fisher), which were then allowed to dry for 1 hr before processing. Cross-sections were incubated overnight with primary antibodies against β-gal (anti-Chicken; 1:4000, Aves) and Myosin-VIIa (anti-Rabbit; 1:500, Proteus BioSciences) for detection of β-gal and with Myosin-VIIa only for qualitative analysis of the *Tecta-Cre;TdT* reporter mouse line. Sections were then rinsed three times with PBS and incubated for two hours at room temperature with secondary antibodies raised in donkey (Alexa-488 and Alexa-546; 1:2000, Life Technologies). Slides were washed three times in PBS (second with PBS + 1:10,000 DAPI), allowed to dry, and sealed using Aqua Polymount (Polysciences, Inc). Images were captured using a laser scanning confocal microscope (LSM 510 or 880, Zeiss).

## Confocal imaging of explants

After one day in vitro, cochleae were moved into a recording chamber and continuously superfused with bicarbonate-buffered artificial cerebrospinal fluid (1.5–2 mL/min) consisting of the following (in mM): 119 NaCl, 2.5 KCl, 1.3 $MgCl_2$, 1.3 $CaCl_2$, 1 $NaH_2PO_4$, 26.2 $NaHCO_3$, 11 D-glucose, and saturated with 95% $O_2$/5% $CO_2$ to maintain a pH of 7.4. A piezo-mounted objective was used to rapidly alternate between SGN cell bodies and ISCs/IHCs. Images were captured at one frame per second using a Zeiss laser scanning confocal microscope (LSM 710, Zeiss) through a 20X objective (Plan APOCHROMAT 20x/1.0 NA) at 512 × 512 pixel (354 × 354 µm; 16-bit depth) resolution. Sections were illuminated with a 488 nm laser (maximum 25 mW power). MRS2500 (1 µM, Tocris) was applied by addition to the superfusing ACSF.

## Analysis of in vitro Ca²⁺ transients

Images were imported into ImageJ and image registration (MultiStackReg) was used to correct for drifts in the imaging field. Since images were obtained at two different z-planes, images were combined into one stack for analysis. This was done by eliminating the empty bottom half of the imaging field containing ISCs and IHCs and the empty top half of the field containing SGN cell bodies and merging the two images. For analysis of coordinated activity throughout the cochlea, regions of interest were drawn around the entirety of ISCs, IHCs, and SGNs. Fluorescence changes were normalized as $\Delta F/F_o$ values, where $\Delta F = F – Fo$ and $F_o$ was defined as the fifth percentile value for each pixel. Peaks in the signals were detected in MATLAB using the built-in peak detection function (findpeaks) with a fixed value threshold criterion (mean + three standard deviations for each cell).

To quantify frequency and areas of Ca²⁺ transients, a threshold of three standard deviations above the mean was applied to each pixel within the ROI. To calculate the frequency of these events, the whole field was taken as an ROI and peaks were detected using MATLAB (findpeaks function) on the number of thresholded pixels per frame. To calculate area of these events, a Gaussian filter (sigma = 2.0) was applied to the image after thresholding and the borders detected using MATLAB (bwlabel function). The area was then calculated as the number of pixels within the border multiplied by an area scaling factor (1 µm/pixel)² measured with a stage micrometer.

For correlation analysis, ROIs were drawn around every IHCs in the field of view. Pairwise correlation coefficients were performed between every hair cell pair and represented as correlation matrices.

## Installation of cranial windows

Inhalation anesthesia was induced with vaporized isoflurane (4% for 5 min, or until mice are non-responsive to toe-pinch) and surgical plane maintained during the procedure (with 1–2% isoflurane) with a stable respiration rate of 80 breaths per minute. A midline incision beginning posterior to the ears and ending just anterior to the eyes was made. Two subsequent cuts were made to remove the dorsal surface of the scalp. A headbar was secured to the head using super glue (Krazy Glue). Fascia and neck muscles overlying the interparietal bone were resected and the area bathed in sterile, HEPES-buffered artificial cerebrospinal fluid that was replaced as necessary throughout the surgery. Using a 28G needle and microblade, the sutures circumscribing the interparietal bone were cut and removed to expose the midbrain. The dura mater was removed using fine scissors and forceps,

exposing the colliculi and extensive vasculature. A 5 mm coverslip (CS-5R; Warner Instruments) was then placed over the craniotomy, the surrounding bone was dried using a Kimwipe, and super glue was placed along the outer edges of the coverslip for adhesion to the skull. Replacement 0.9% NaCl solution was injected IP and a local injection of lidocaine was given to the back of the neck. Animals were weaned off isoflurane, placed under a warming lamp, and allowed to recover for a minimum of 1 hr prior to imaging. Spontaneous activity was not seen in deeply anesthetized animals and emerged ~30 min after recovery from isoflurane exposure, as reported previously (*Ackman et al., 2012*).

## In vivo calcium imaging

After 1 hr of post-surgical recovery from anesthesia, pups were moved into a swaddling 15 mL conical centrifuge tube. The top half of this tube was removed to allow access to the headbar and visualization of the midbrain. Pups were head-fixed and maintained at 37°C using a heating pad and temperature controller (TC-1000; CWE). During the experiments, pups were generally immobile; however, occasional limb and tail twitching did occur.

For wide field epifluorescence imaging, images were captured at 10 Hz using a Hamamatsu ORCA-Flash4.0 LT digital CMOS camera attached to a Zeiss Axio Zoom.V16 stereo zoom microscope. A $4 \times 4$ mm field of view was illuminated continuously with a metal halide lamp (Zeiss Illuminator HXP 200C) and visualized through a 1X PlanNeoFluar Z 1.0x objective at 17x zoom. Images were captured at a resolution of $512 \times 512$ pixels (16-bit pixel depth) after $2 \times 2$ binning to increase sensitivity. Each recording consisted of uninterrupted acquisition over 10 min or 20 min if injected with pharmacological agents.

## Catheterization of animals for in vivo imaging

After induction of anesthesia and before installing the cranial window, a catheter was placed in the intraperitoneal (IP) space of neonatal mouse pups. A 24G needle was used to puncture the peritoneum and a small-diameter catheter (SAI Infusion Technologies, MIT-01) was placed. A drop of Vetbond secured the catheter to the pup's belly. Installation of cranial window proceeded as described above.

Imaging sessions consisted of 5 min of baseline activity measurements, followed by a slow push of either 50 µL of sham (5% mannitol solution) or MRS2500 solution (500 µM in 5% mannitol solution). Imaging was continuous throughout and 20 min of activity total were collected. No discernable diminishment of activity was observed in sham animals.

## Image processing

For wide field imaging, raw images were imported into the ImageJ environment and corrected for photobleaching by fitting a single exponential to the fluorescence decay and subtracting this component from the signal (Bleach Correct function, exponential fit). Images were then imported into MATLAB (Mathworks) and intensities were normalized as $\Delta F/F_o$ values, where $\Delta F = F - F_o$ and $F_o$ was defined as the fifth percentile value for each pixel. Ovoid regions of interest (ROIs) encompassing the entire left and right inferior colliculi were drawn. Across all conditions, the size of the ROIs was invariant, however, due to small differences in the imaging field between animals, the ROIs were placed manually for each imaging session. Peaks in the signals were detected in MATLAB using the built-in peak detection function (findpeaks) using a fixed value threshold criterion; because fluorescence values were normalized, this threshold was fixed across conditions (2% $\Delta F/F_o$). Occasionally, large events in the cortex or superior colliculus would result in detectable fluorescence increases in the IC. These events broadly activated the entire surface of the IC and did not exhibit the same spatially-confined characteristics as events driven by the periphery. These events were not included in the analysis.

## Analysis of spatial distribution of activity in the IC

As shown in *Figure 6D*, a rectangle of size $125 \times 50$ pixels was placed perpendicular to the tonotopic axis of the IC (±55° rotation, respectively). The columns of the resulting matrix were averaged together to create a line scan (125 pixels x one pixel) for the entire time series. Peaks were detected using MATLAB's imregionalmax function with a constant threshold of 3% $\Delta F/F_o$ across all animals.

Histograms of events along the tonotopic axis were generated by summing the number of events in ~25 μm bins. Lateral and medial designations were assigned by splitting the area evenly between the lateral edge and the location of defined single-band events in the medial portion of the IC. Events detected on the medial edge of single-band events, reflective of the bifurcation of this information was not included in the medial/lateral analysis.

### Analysis of retinal wave activity in the superior colliculus

ROIs (200 × 150 pixels) were placed over each lobe of the superior colliculus and downsampled by a factor of five. Signals were normalized as $\Delta F/F_o$ values, where $\Delta F = F - F_o$ and $F_o$ was defined as the fifth percentile value for each pixel. In order to eliminate periodic whole-sample increases in fluorescence, the mean intensity of all pixels was subtracted from each individual pixel. Following this, pixels were considered active if they exceeded the mean + three standard deviations. For each point in time, the number of active pixels was summed. Retinal waves were defined as prolonged periods (>1 s), where more than five pixels were active simultaneously. Retinal wave durations were defined as the total continuous amount of time that more than five pixels were active. Frequencies and durations are similar to earlier reports (*Ackman et al., 2012*).

### Generation of the Tecta-Cre mouseline

A crRNA (TAATGATGAATAATTCATCC) targeted near exon 2 of the *Tecta* gene, tracrRNA, Cas9 recombinase, and a donor plasmid containing an iCre-WPRE-polyA sequence (500 base pair homology arms) were injected into single-cell embryos that were then transferred to pseudopregnant recipient mothers. After birth, mouse pups were screened for insertion of the gene at the correct locus with two pairs of primers: one pair amplified DNA beginning 5' of the 5' homology arm and ending within the Cre sequence and the other amplified DNA within the polyA sequence and ending 3' of the 3' homology arm. These primers were then used to sequence the junctions. Of these, all mice used for experiments were derived from a single founder that was positive for both sets of primers and had 100% sequence validation. Mice were crossed to a TdTomato reporter line to examine cell-specific recombination.

### Quantification and statistical analysis

All statistics were performed in the MATLAB (Mathworks) programming environment. All statistical details, including the exact value of n, what n represents, and which statistical test was performed, can be found in the figure legends. To achieve statistical power of 0.8 with a 30% effect size with means and standard deviations similar to those observed in previous studies (Figure 1E of *Tritsch et al., 2007* and Figures 1B and 3D in *Wang and Bergles, 2015*), power calculations indicated that seven animals in each condition are necessary ($\mu_1 = 10$, $\mu_2 = 7$, $\sigma = 2$, sampling ratio = 1). While this number was used as a guide, power calculations were not explicitly performed before each experiment; many experiments had much larger effect sizes and sample sizes were adjusted accordingly. For transparency, all individual measurements are included in the figures. Unless otherwise noted, data are presented as mean ± standard error of the mean (SEM). Because the main comparison between conditions was the mean, the SEM is displayed to highlight the dispersion of sample means around the population mean. All datasets were tested for Gaussian normality using the D'Agostino's $K^2$ test. For single comparisons, significance was defined as $p<=0.05$. When multiple comparisons were made, the Bonferroni correction was used to adjust p-values accordingly to lower the probability of type I errors. For multiple condition datasets, one-way ANOVAs were used, followed by Tukey's multiple comparison tests.

## Acknowledgements

We thank Dr. M Pucak and N Ye for technical assistance, T Shelly for machining expertise, members of the Bergles laboratory and Dr. Ulrich Müller for discussions and comments on the manuscript. TB was supported by a NRSA grant from the NIH (F31DC016497). Funding was provided by grants from the NIH (DC008060, NS050274), Otonomy Inc, the Brain Science Institute at Johns Hopkins University, and a grant from the Rubenstein Fund for Hearing Research.

# Additional information

## Competing interests

Dwight E Bergles: Reviewing editor, *eLife*. The other authors declare that no competing interests exist.

## Funding

| Funder | Grant reference number | Author |
|---|---|---|
| National Institute on Deafness and Other Communication Disorders | DC016497 | Travis A Babola |
| National Institute on Deafness and Other Communication Disorders | DC008860 | Dwight E Bergles |
| National Institute of Neurological Disorders and Stroke | NS091018 | Travis A Babola |
| National Institute on Deafness and Other Communication Disorders | DC000023 | Travis A Babola |
| Brain Science Institute, Johns Hopkins University | | Dwight E Bergles |
| Rubenstein Fund for Hearing Research | | Dwight E Bergles |
| Otonomy Inc | | Dwight E Bergles |
| National Institute of Neurological Disorders and Stroke | NS050274 | Dwight E Bergles |

The funders had no role in study design, data collection and interpretation, or the decision to submit the work for publication.

## Author contributions

Travis A Babola, Conceptualization, Data curation, Software, Formal analysis, Funding acquisition, Validation, Investigation, Visualization, Methodology; Calvin J Kersbergen, Data curation, Formal analysis, Validation, Investigation, Methodology; Han Chin Wang, Data curation, Formal analysis, Investigation, Methodology; Dwight E Bergles, Conceptualization, Data curation, Supervision, Funding acquisition, Methodology, Project administration

## Author ORCIDs

Travis A Babola  https://orcid.org/0000-0003-4440-5029
Dwight E Bergles  https://orcid.org/0000-0002-7133-7378

## Ethics

Animal experimentation: This study was performed in accordance with the recommendations provided in the Guide for the Care and Use of Laboratory Animals of the National Institutes of Health. All experiments and procedures were approved by the Johns Hopkins Institutional Care and Use Committee (protocol M018M330). All surgery was performed under isoflurane anesthesia and every effort was made to minimize suffering.

## Decision letter and Author response

Decision letter https://doi.org/10.7554/eLife.52160.sa1
Author response https://doi.org/10.7554/eLife.52160.sa2

## Additional files

### Supplementary files
• Transparent reporting form

### Data availability

All data generated or analyzed in this study are included in the manuscript. Source code for analysis and figure generation are located at: https://github.com/tbabola/P2ry1_eLife_SourceCode (copy archived at https://github.com/elifesciences-publications/P2ry1_eLife_SourceCode).

The following previously published dataset was used:

| Author(s) | Year | Dataset title | Dataset URL | Database and Identifier |
|---|---|---|---|---|
| Scheffer DI, Shen J, Corey DP, Chen Z | 2015 | Gene Expression by Mouse Inner Ear Hair Cells During Development | https://www.ncbi.nlm.nih.gov/geo/query/acc.cgi?acc=GSE60019 | NCBI Gene Expression Omnibus, GSE60019 |

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
