## [Decision Letter]

**Acceptance summary:**

This study employs an impressive combination of methods to investigate the mechanisms that trigger spontaneous patterned activity in mouse inner hair cells prior to hearing onset. This spontaneous activity is conveyed along the auditory pathway and is thought to play an important role in the development of the auditory system. Building on previous work by the authors showing that ATP release is involved in this process, the present study uses pharmacological tools and knockout mice to demonstrate that purinergic signaling in supporting cells, mediated by P2RY1 autoreceptors, regulates hair cell excitability by controlling the volume of the extracellular space. Clear evidence is presented that this results from crenation – osmotic shrinkage – of the supporting cells, which increases the extracellular space. This study therefore provides insight into the mechanisms that initiate intrinsically generated bursts of activity in the developing auditory system, abnormalities of which may contribute to certain developmental hearing disorders.

**Decision letter after peer review:**

Thank you for submitting your article "Supporting cells in the cochlea reduce hair cell excitability by increasing the extracellular space" for consideration by *eLife*. Your article has been reviewed by Andrew King as the Senior Editor and Reviewing Editor, and three reviewers. The following individuals involved in review of your submission have agreed to reveal their identity: Ivan Milenkovic (Reviewer #1); Christian Lohr (Reviewer #2).

The reviewers have discussed the reviews with one another and the Reviewing Editor has drafted this decision to help you prepare a revised submission.

The reviewers were all enthusiastic about the paper, which they felt was appropriate for *eLife*. Although they did not indicate that additional experimental work is essential, a concern was raised about the discrepancy between the reported effect of UTP (agonist at P2Y2, P2Y4 and P2Y6 receptors) in recent publications and the apparently exclusive effect of P2Y1 ligands in the present manuscript. This issue needs to be addressed, if necessary, with additional data. Furthermore, the reviewers all felt that some of the findings have been over-interpreted and that the paper is currently difficult to follow due to the wealth of manipulations and different approaches used.

Essential revisions:

1) One of the main conclusions is that P2Y1-induced SC crenation is temporally matched by the shift in the IHC potassium current (Figure 4). This is assumed to change the burst structure recorded in spiral ganglion neurons (Figure 6). Indeed, the 75-125 ms interspike intervals are strongly reduced, while the remaining ones are shifted to longer values. The time constant for crenations is about 40 ms, and 100 ms for the potassium current. MRS2500 completely blocks spontaneous currents in the IHCs before the late inward current arises (Figure 3F,G). How does MRS2500 eliminate the whole bursts and the spikes within a burst in Figure 6A-F? Moreover, the spiking under MRS2500 is transiently abolished prior to firing at even higher rate, presumably due to increased excitability. These temporal relationships between the crenations, changes in potassium current direction, and the loss of interspike intervals should be better elaborated and discussed.

2) Although the MRS2500 injection experiments in Figure 8 show a profound reduction of activity in the IC, it is not possible to exclude the possibility of P2Y1 inhibition elsewhere in the auditory pathway. For instance, cochlear nucleus bushy cells express P2Y1 (Milenkovic et al., 2009) and it is conceivable that some other neurons may express them too. The assumption that MRS2500 crosses the blood-brain barrier only in the cochlea seems unjustified. Due to the inaccessibility of the cochlea for a local drug application, the authors performed what is feasible, but they should clearly discuss that this may not be a cochlea-specific effect.

3) In their recent publications (Tritsch and Bergles, 2010; Wang et al., 2015), the authors showed that UTP is as potent as ATP in triggering the signaling cascade that results in IHC depolarization. UTP is not a potent agonist at P2Y1 receptors, but rather stimulates P2Y2, P2Y4 and to some extent P2Y6, suggesting that several P2Y receptor subtypes might be involved. It is surprising that the authors have not even mentioned this discrepancy. They show that selective stimulation of P2Y1 with MRS2365 fails to induce currents and crenation in inner supporting cells of P2Y1 KO mice, which is expected given that MRS2365 is selective for P2Y1. Is the UTP-evoked effect also suppressed in P2Y1 KO mice (or by blocking P2Y1 with MRS2500 in wildtype mice)? Spontaneous activity in inner supporting cells and IHCs is entirely suppressed by the P2Y1 antagonist MRS2500 with no further effect of suramin/PPADS, suggesting no major contribution of other P2Y receptors. What is the role of UTP-sensitive P2Y receptors in purinergic activity in the developing cochlea? This issue should at least be discussed.

4) The reviewers disagreed with the assertion in the subsection “Supporting cell spontaneous currents require calcium release from intracellular stores” that U73122 has a dramatic effect on the frequency of spontaneous currents and crenation. In Figure 1D,E, the effect is not even significant compared to U73343, which was used as negative control. The lack of a complete block by U73122 (which would be expected at the concentration of 10µM), and no difference in the effect between the blocker and the inactive analogue, raise the question of whether calcium is released from the ER solely via IP3. In line with this, neither BAPTA nor Thapsigargin are specific for the Gq pathway, as stated in the legend for Figure 1C. Both drugs would have the same effect if the calcium is released via RyRs. Thus, the conclusions about the sole contribution of the Gq pathway may not be fully justified.

5) Statistics: Please explain the power calculation more thoroughly at least for one example. From what type of data in the literature do you expect a 30% effect size and 20% SD?

6) The manuscript uses SEM in the figures. The choice of SEM over SD should be justified, and all the data points shown (this is done for most but not all panels). The number of animals in addition to number of cells should be reported.

7) Figures: in some cases, it is difficult to find the age of the animal for a given figure panel. Please specify this in all figure legends.

8) Differences in Figure 7 and Figure 8 are in some cases subtle or not significant, but these results are underemphasized in the overall interpretation of the results. The Introduction sets up the problem in an overly dichotomous perspective regarding importance of different mechanisms for generating spontaneous activity. A more helpful treatment would consider both ISC activity and efferent activity as essential parts of the overall process.

9) The Discussion section should consider how the whole cycle might be regulated at the macroscopic level. What initiates release of ATP from the supporting cells?

10) Possible CNS effects on results in KO mice are considered, but the logic of these needs to be explained a bit more.

---

## [Author Response]

Essential revisions:1) One of the main conclusions is that P2Y1-induced SC crenation is temporally matched by the shift in the IHC potassium current (Figure 4). This is assumed to change the burst structure recorded in spiral ganglion neurons (Figure 6). Indeed, the 75-125 ms interspike intervals are strongly reduced, while the remaining ones are shifted to longer values. The time constant for crenations is about 40 ms, and 100 ms for the potassium current. MRS2500 completely blocks spontaneous currents in the IHCs before the late inward current arises (Figure 3F,G). Moreover, the spiking under MRS2500 is transiently abolished prior to firing at even higher rate, presumably due to increased excitability.How does MRS2500 eliminate the whole bursts and the spikes within a burst in Figure 6A-F?

Application of MRS2500 blocks P2RY1-mediated spontaneous currents (Figure 3B,F), eliminating large transient increases in extracellular K^+^ (Figure 4G) and therefore IHC burst firing (Figure 6D). Just as activation of P2RY1 increases the extracellular space by causing supporting cell shrinkage, inhibition of P2RY1 causes supporting cells to expand, decreasing the extracellular space. This decrease in the extracellular volume leads to accumulation of K^+^ (Figure 3B,H), resulting in slow depolarization (Figure 3F) and tonic (non-burst) firing of IHCs (Figure 6D). Thus, P2RY1 antagonism shifts SGNs from a burst firing mode to random firing, leading to loss of bursts and spikes within a burst.

These temporal relationships between the crenations, changes in potassium current direction, and the loss of interspike intervals should be better elaborated and discussed.

To clarify the temporal relationship between these events more clearly, we now include a summary figure detailing the time course of these discrete events (Figure 9), and have expanded the discussion of the sequence of changes that lead to the shift in firing behavior of SGNs when P2RY1 is inhibited (subsection “Control of extracellular K^+^ dynamics by supporting cells”).

2) Although the MRS2500 injection experiments in Figure 8 show a profound reduction of activity in the IC, it is not possible to exclude the possibility of P2Y1 inhibition elsewhere in the auditory pathway. For instance, cochlear nucleus bushy cells express P2Y1 (Milenkovic et al., 2009) and it is conceivable that some other neurons may express them too. The assumption that MRS2500 crosses the blood-brain barrier only in the cochlea seems unjustified. Due to the inaccessibility of the cochlea for a local drug application, the authors performed what is feasible, but they should clearly discuss that this may not be a cochlea-specific effect.

We agree that MRS2500 may also act on P2RY1 receptors in the CNS, and did not mean to imply that this compound only has access to the cochlea. The preservation of retina-induced activity in the SC (Figure 8) after MRS2500 administration suggests that global changes in neuronal activity are not affected by this drug; however, it is possible that there may be specific roles for P2RY1 in the auditory pathway, as *P2ry1* mRNA is expressed in the cochlear nucleus and P2RY1-mediated calcium transients can be induced in spherical bushy cells (Milenkovic et al., 2009). At present, we do not know the conditions under which these receptors would be activated or what effect their inhibition would have on the propagation of bursts of activity to the IC. The discussion of CNS P2RY1 receptors has been expanded to address this possibility (subsection “Role of supporting cells in the generation of spontaneous activity”).

3) In their recent publications (Tritsch and Bergles, 2010; Wang et al., 2015), the authors showed that UTP is as potent as ATP in triggering the signaling cascade that results in IHC depolarization. UTP is not a potent agonist at P2Y1 receptors, but rather stimulates P2Y2, P2Y4 and to some extent P2Y6, suggesting that several P2Y receptor subtypes might be involved. It is surprising that the authors have not even mentioned this discrepancy. They show that selective stimulation of P2Y1 with MRS2365 fails to induce currents and crenation in inner supporting cells of P2Y1 KO mice, which is expected given that MRS2365 is selective for P2Y1. Is the UTP-evoked effect also suppressed in P2Y1 KO mice (or by blocking P2Y1 with MRS2500 in wildtype mice)? Spontaneous activity in inner supporting cells and IHCs is entirely suppressed by the P2Y1 antagonist MRS2500 with no further effect of suramin/PPADS, suggesting no major contribution of other P2Y receptors. What is the role of UTP-sensitive P2Y receptors in purinergic activity in the developing cochlea? This issue should at least be discussed.

The reviewers raise an important issue regarding the complement of different P2Y receptors expressed by supporting cells in the cochlea. We did not directly address this question in this study and our experiments do not rule out the possibility that supporting cells express other P2Y receptors. Our focus was to identify which purinergic receptors are responsible for spontaneous activity. The SHIELD gene expression database reveals the mRNAs for *P2ry1, P2ry2* and *P2ry4* G_q_-coupled receptors are present, although *P2ry2* and 4 mRNA are present at much lower levels than *P2ry1* (100x less, Figure 2A, Scheffer et al., 2015). Our experiments demonstrate that P2RY1 is both necessary and sufficient to generate transient inward currents in supporting cells. Applying a P2RY1-specific agonist, MRS2365, generates large inward currents (Figure 4B,C) and applying a P2RY1 antagonist, MRS2500, eliminates large spontaneous inward currents (Figure 3B,C), an effect also observed in *P2ry1* KO mice (Figure 3—figure supplement 2). Additionally, large calcium transients are abolished in supporting cells by acute P2RY1 inhibition or genetic deletion (Figure 5 and Figure 5—figure supplement 1). Together, these data suggest that, if other P2Y autoreceptors are expressed, they are not principally involved in generating spontaneous activity.

Our prior studies and gene expression analyses suggest that inner supporting cells express a variety of functional P2X and P2Y receptors, which can be activated by exogenous agonists. We do not know why these other receptors are not activated in inner supporting cells during spontaneous events. It is possible that the ATP is rapidly hydrolyzed to ADP by extracellular nucleotidases, limiting activation of P2X receptors, and the much greater abundance of P2RY1 may enable preferential activation of these receptors, particularly if the amount of ATP released is low. Further in vivogenetic and pharmacological manipulations will be required to define the conditions under which these receptors are activated and their consequences for spontaneous and evoked activity. We have expanded discussion of these issues (subsection “Control of extracellular K^+^ dynamics by supporting cells” and subsection “Purinergic receptors in the adult cochlea”).

4) The reviewers disagreed with the assertion in subsection “Supporting cell spontaneous currents require calcium release from intracellular stores” that U73122 has a dramatic effect on the frequency of spontaneous currents and crenation. In Figure 1D,E, the effect is not even significant compared to U73343, which was used as negative control. The lack of a complete block by U73122 (which would be expected at the concentration of 10µM), and no difference in the effect between the blocker and the inactive analogue, raise the question of whether calcium is released from the ER solely via IP3. In line with this, neither BAPTA nor Thapsigargin are specific for the Gq pathway, as stated in the legend for Figure 1C. Both drugs would have the same effect if the calcium is released via RyRs. Thus, the conclusions about the sole contribution of the Gq pathway may not be fully justified.

As noted, the PLC inhibitor did not abolish spontaneous currents in supporting cells, raising the possibility that there are other mechanisms responsible for the spontaneous calcium transients. The significant decrease in frequency of inward currents and crenations by U73211 (Figure 1C,F) supports the involvement of PLC, and is consistent with the canonical signal transduction mechanism expected for P2RY1; however, it is possible that calcium stores could be mobilized by activation of ryanodine receptors (RyRs) via other purinergic receptors. To explore this possibility, we examined the effect of acute exposure to ryanodine (10 µM) on ISCs. This manipulation did not decrease either spontaneous inward currents or crenations, which is now included as a new figure (Figure 1—figure supplement 1) and described in subsection “The metabotropic purinergic receptor P2Y1 is highly expressed by supporting cells”. These findings are consistent with information from the SHIELD database, which suggests that RyRs are not expressed highly by supporting cells (*Ryr1*: 1-4 normalized reads between E16 and P7, *Ryr2*: 2-44 normalized reads, *Ryr3*: 0-7 normalized reads, *P2ry1*: 1010-1742 normalized reads; Scheffer et al., 2015).

We revised the Figure legend text (1C) to remove references to the G_q_ pathway.

5) Statistics: Please explain the power calculation more thoroughly at least for one example. From what type of data in the literature do you expect a 30% effect size and 20% SD?

References to previous studies using similar techniques are now included in the subsection “Quantification and Statistical Analysis”. In Figure 1E of Tritsch et al., 2007 and in Figures 1B and 3D in Wang et al., 2015, standard deviations range from ~15 to 25%. A 30% effect size was selected as a conservative estimate. Our previous studies have shown that effect sizes when spontaneous activity is pharmacologically or genetically altered is much higher (between 50 and 90%; Tritsch et al., 2007, Wang et al., 2015).

6) The manuscript uses SEM in the figures. The choice of SEM over SD should be justified, and all the data points shown (this is done for most but not all panels). The number of animals in addition to number of cells should be reported.

Justification for the use of SEM is now included in subsection “Quantification and Statistical Analysis*”*. Briefly, SEM was chosen as it shows the dispersion of sample means around the population mean. Because the means of measurements are being compared across conditions, we feel that the SEM provides more valuable information for visual comparison than the SD, which provides information about the variability of the sample. All individual data points are included, and the number of animals has been added to the figure legends.

7) Figures: in some cases, it is difficult to find the age of the animal for a given figure panel. Please specify this in all figure legends.

All experiments were performed in P6-8 animals. This information has been added to the figure legends.

8) Differences in Figure 7 and Figure 8 are in some cases subtle or not significant, but these results are underemphasized in the overall interpretation of the results. The Introduction sets up the problem in an overly dichotomous perspective regarding importance of different mechanisms for generating spontaneous activity. A more helpful treatment would consider both ISC activity and efferent activity as essential parts of the overall process.

We observed differences in *P2ry1* KO mice and the response to acute inhibition of P2RY1. The effects of the acute P2RY1 inhibition, both in excised cochlea (Figure 3, Figure 5, Figure 6) and in vivo (Figure 8) are highly significant. However, differences between control and *P2ry1* KO mice were, in some cases, subtle (Figure 7), as noted. We do not yet know the explanation for these differences but expect that constitutive loss of P2RY1 throughout development leads to compensatory alterations. Indeed, our studies indicate that IHCs are more depolarized in *P2ry1* KO mice, suggesting that they may be more sensitive to other excitatory influences. While we did not directly address the role of efferents in generating spontaneous activity, our excised cochleae do not have CNS input, recent in vivo recordings from *alpha9* KO mice (Clause et al., 2014) and our own in vivo imaging studies from these animals (unpublished) indicate that spontaneous activity persists in the absence of efferent cholinergic input to IHCs. We did not intend to imply that efferents play no role in spontaneous activity and now more explicitly define the potential roles of supporting cells and efferents in the Introduction and subsection “Purinergic signaling in the developing cochlea”).

9) The Discussion section should consider how the whole cycle might be regulated at the macroscopic level. What initiates release of ATP from the supporting cells?

Unfortunately, we do not yet know how ATP is released by supporting cells. The available data suggest that a gap-junction hemichannel might be responsible, as gap-junction inhibitors profoundly inhibit spontaneous activity, and lowering extracellular calcium, a manipulation that increases the open probability of hemi-channels (Peracchia, 2004), enhances the frequency of spontaneous activity (Tritsch et al., 2007). Moreover, our unpublished results indicate that inhibiting vesicular loading with bafilomycin-A does not alter spontaneous activity in supporting cells. However, due to cochlear expression of many gap-junction family members and other potential ATP-release channels (VRACs, maxi-anion, pannexins) (Lautermann et al., 1998, Zhang et al., 2005, Scheffer et al., 2015) and the absence of specific pharmacological agents to inhibit these channels, it has not yet been possible to explore this question further. We now address this issue briefly in subsection “Purinergic signaling in the developing cochlea”.

10) Possible CNS effects on results in KO mice are considered, but the logic of these needs to be explained a bit more.

It is difficult to predict what compensatory changes might occur without P2RY1, as little is known about the role of P2RY1 in the CNS or even the broader role of purinergic receptors in CNS development. The most notable effect of P2ry1 KO is the depolarization of IHCs, which may enhance the effect of previously minor/subthreshold events. Enhanced, non-purinergic spontaneous activity is also evident in supporting cells from *P2ry1* KO mice, raising the possibility of other gain-of-function changes. These changes are now discussed (subsection “Purinergic receptors in the adult cochlea”).